# MixLinear: Extreme Low Resource Multivariate Time Series Forecasting with $0.1K$ Parameters

**Aitian Ma, Dongsheng Luo, Mo Sha**[*]
Knight Foundation School of Computing and Information Sciences
Florida International University
{ama003, dluo, msha}@fiu.edu

## Abstract

Recently, there has been a growing interest in Long-term Time Series Forecasting (LTSF), which involves predicting long-term future values by analyzing a large amount of historical time-series data to identify patterns and trends. Significant challenges exist in LTSF due to its complex temporal dependencies and high computational demands. Although Transformer-based models offer high forecasting accuracy, they are often too compute-intensive to be deployed on devices with hardware constraints. In this paper, we propose MixLinear, which synergistically combines segment-based trend extraction in the time domain with adaptive low-rank spectral filtering in the frequency domain. Our approach exploits the complementary structural sparsity of time series: local temporal patterns are efficiently captured through mathematically linear transformations that separate intra-segment and inter-segment correlations, while global trends are compressed into an ultra-low-dimensional frequency latent space through learnable rank-constrained filters. By reducing the parameter scale of a downsampled $n$-length input/output one-layer linear model from $O(n^2)$ to $O(n)$, MixLinear achieves efficient computation without sacrificing accuracy. Extensive evaluations show that MixLinear achieves forecasting performance comparable to existing models with significantly fewer parameters $(0.1K)$, which makes it well-suited for deployment on devices with limited computational capacity.

## 1 Introduction

Deep learning models have recently achieved state-of-the-art accuracy on time series forecasting in various applications (Moon & Wettlaufer, 2017; Nunnari & Nunnari, 2017; Sezer et al., 2020; Ma et al., 2025c). However, this success is marked by a trend of escalating computational cost and parameter counts, mirroring the trajectory of models in NLP and vision (Brown et al., 2020; Kaplan et al., 2020; Dosovitskiy et al., 2021; Radford et al., 2021). With leading models comprising millions of parameters, their deployment on embedded devices, edge sensors, and other resource-constrained systems is often infeasible. We argue this parameter explosion is not an inevitable price for performance but a symptom of a structural inefficiency in how current models represent time series patterns (Zhou et al., 2022b; Wu et al., 2021; 2023; Xu et al., 2024).

The core of this inefficiency lies in a monolithic representational strategy. Prevailing architectures attempt to capture both high-frequency local variations and low-frequency global patterns using the same set of mechanisms, despite their inherently different statistical properties. Local features (e.g., short-term fluctuations) are best characterized by their temporal locality, while global structures (e.g., long-term trends and seasonalities) are well-known to be sparse in the frequency domain (Zhou et al., 2022b; Yi et al., 2023). Forcing a single, uniform architecture to model these disparate characteristics is a mismatch that leads directly to parameter redundancy and computational waste.

Recent research has started to process local and global components differently. One approach, found in models like DeepGate (Park et al., 2022), decomposes the time series first. However, such a method

---

[*]✉ Corresponding author.

often still relies on complex, parameter-heavy modules for both components, failing to achieve the desired level of efficiency. Another approach uses the frequency domain. Models like FITS (Xu et al., 2024) are highly efficient for global patterns. However, using global frequency components to model localized, transient variations is inherently inefficient; it requires a disproportionate number of coefficients to capture sharp features that would be simple to represent in the time domain. This diminishes the efficiency gains from spectral compression. Therefore, the central challenge is to design a framework that can effectively model both global and local patterns while remaining computationally efficient.

Our work is motivated by a core insight about time series patterns: *local trends can be efficiently captured through segment-based extraction in the time domain, while global trends exhibit sparsity in the frequency domain and benefit from adaptive low-rank filtering*. This insight suggests that optimal efficiency requires not just separating patterns, but processing each in its most natural domain—time for local trends and frequency for global trends. To realize this vision, we introduce MixLinear, a dual-pathway framework that systematically processes local trends through factorized linear decomposition in the time domain and global trends through adaptive low-rank spectral filtering in the frequency domain. MixLinear addresses three fundamental challenges: (1) developing parameter-efficient segment-based extraction that reduces complexity from $\mathcal{O}(n^2)$ to $\mathcal{O}(n)$ (for an original series length $L$ and downsampling factor $\pi$, $n = L/\pi$) while preserving hierarchical temporal structures, (2) achieving extreme compression through rank-constrained spectral filtering that focuses on dominant frequency modes, and (3) integrating dual pathways through learnable upsampling without parameter explosion.

Our contributions are as follows:

- We introduce an efficient approach for local trend processing that reduces complexity from $\mathrm{O}(n^2)$ to $\mathrm{O}(n)$ through strategic segmentation and dual linear transformations for intra-segment and inter-segment dependencies.
- We propose a complex-valued transformation approach with adaptive low-rank filtering that compresses global trends into a minimal latent space, achieving unprecedented parameter efficiency while preserving essential spectral information.
- We create a novel synthesis that seamlessly combines time-domain and frequency-domain processing to achieve an unprecedented operating point on the efficiency-accuracy curve.
- Extensive experiments on benchmark datasets demonstrate that MixLinear achieves competitive forecasting performance with remarkable efficiency — orders of magnitude fewer parameters than existing lightweight models. Our work opens new possibilities for forecasting on resource-constrained devices and provides a practical framework for efficient time series representation learning.

## 2 MixLinear Design

### 2.1 Preliminary

LTSF involves predicting future values over an extended horizon using previously observed multivariate time series data. It is formalized as $\hat{x}_{t+1:t+H} = f(x_{t-L+1:t})$, where $x_{t-L+1:t} \in \mathbb{R}^{L \times C}$ and $\hat{x}_{t+1:t+H} \in \mathbb{R}^{H \times C}$. In this formulation, $L$ denotes the length of the historical observation window, $C$ represents the number of distinct features or channels, and $H$ denotes the length of the forecast horizon. Extending the forecast horizon $H$ is crucial for LTSF as it allows more comprehensive long-term planning and decision making in practical applications. However, increasing the forecast horizon $H$ usually requires more parameters and significantly increases the computational complexity of the forecasting model. Efficiently balancing extended forecast horizons with model complexity remains a key challenge in LTSF research, necessitating novel architectural designs and parameter-efficient strategies to maintain both accuracy and scalability.

### 2.2 Framework Overview

We propose MixLinear, a dual-domain architecture that fundamentally reconceptualizes temporal pattern modeling through factorized representational learning. Our theoretical contribution rests on the *Spectral-Temporal Decomposition Principle*: real-world time series exhibit intrinsic structural duality —— local dynamics manifest as segment-wise correlations amenable to factorized linear

Segment-based Trend Extraction

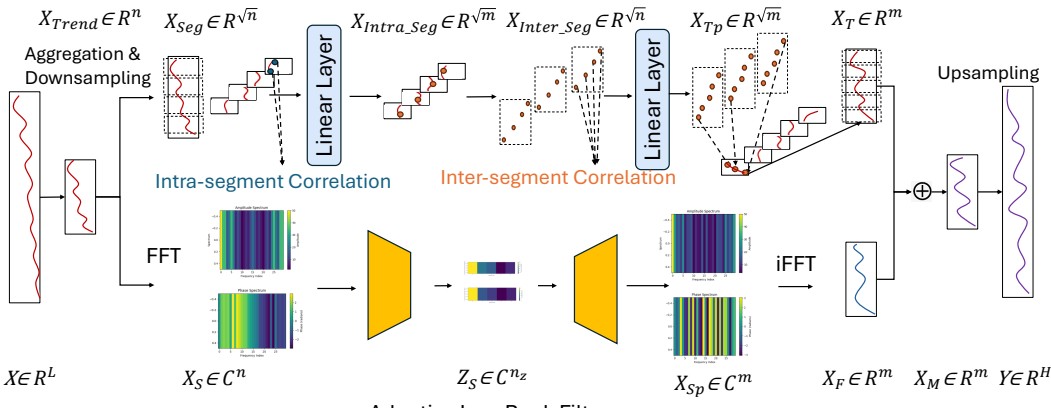

Adaptive Low Rank Filter-based Frequency Domain Trend Extraction

Figure 1: **MixLinear Architecture Overview.** Our dual-pathway framework efficiently processes time series data. The **Segment-based pathway** (top) downsamples input $X \in \mathbb{R}^L$ into segments $X_{seg} \in \mathbb{R}^{L/\pi}$, applies linear transformations for intra-segment (blue) and inter-segment (orange) correlations, then upsamples to $X_T \in \mathbb{R}^H$. The **Frequency-domain pathway** (bottom) transforms segments via FFT ($X_S \in \mathbb{C}^{L/\pi}$), compresses trends through adaptive low-rank filtering to latent space $Z_S \in \mathbb{C}^{n_z}$, reconstructs via iFFT, and outputs $X_F \in \mathbb{R}^H$. Final predictions $Y \in \mathbb{R}^H$ combine both outputs, achieving competitive forecasting with only 0.1K parameters.

decomposition, while global patterns demonstrate spectral concentration enabling extreme low-rank compression in the frequency domain (Zhou et al., 2022b).

MixLinear operates through a sophisticated dual-pathway architecture that synergistically processes complementary temporal manifolds. Given input $\mathbf{X} \in \mathbb{R}^{L \times C}$, our framework produces predictions through the learned compositional mapping:

$$\mathbf{Y} = \mathcal{F}_{\text{segment}}(\mathbf{X}; \boldsymbol{\Theta}_s) + \mathcal{F}_{\text{frequency}}(\mathbf{X}; \boldsymbol{\Theta}_f), \tag{1}$$

where $\mathcal{F}_{\text{segment}}$ performs segment-based trend extraction with parameters $\boldsymbol{\Theta}_s$, and $\mathcal{F}_{\text{frequency}}$ performs adaptive low-rank spectral filtering with parameters $\boldsymbol{\Theta}_f$. This additive composition preserves the independence of domain-specific representations while enabling joint optimization through backpropagation, contrasting with multiplicative fusion approaches that suffer from gradient instability (Vaswani et al., 2017).

The architecture achieves parameter efficiency (0.1K parameters) through principled exploitation of inherent structural sparsity: factorized linear projections capture segment-wise correlations (Kitaev et al., 2020), while rank-constrained spectral filtering enables extreme compression of global modes (Tay et al., 2020). This decomposition fundamentally avoids the parameter explosion endemic to hybrid time-frequency models (Zhou et al., 2021; Liu et al., 2021).

### 2.3 SEGMENT-BASED TREND EXTRACTION

The segment-based pathway introduces a *Factorized Linear Decomposition* framework that disentangles intra-segment and inter-segment correlations through specialized linear projections. This approach transcends traditional segmentation methods (Zeng et al., 2023) by introducing transformations that separate local shape modeling from global trend aggregation.

We implement a principled temporal decomposition strategy through adaptive downsampling and non-overlapping segmentation. Given input $\mathbf{X} \in \mathbb{R}^{L \times C}$, we first apply downsampling by factor $\pi$ to obtain $\mathbf{X}_{\text{down}} \in \mathbb{R}^{(L/\pi) \times C}$, implementing implicit low-pass filtering that attenuates high-frequency noise while preserving trend information (Wu et al., 2021; Cleveland et al., 1990). The downsampled sequence undergoes structured partitioning into $M$ segments of length $r = L/(\pi \cdot M)$:

$$\mathbf{X}_{\text{seg}} = \{\mathbf{x}^{(s)} \in \mathbb{R}^{r \times C}\}_{s=1}^M \tag{2}$$

This decomposition enables parallel processing of local patterns while maintaining computational tractability, analogous to efficient attention mechanisms (Wang et al., 2020; Choromanski et al., 2021). Within each segment $\mathbf{x}^{(s)} \in \mathbb{R}^{r \times C}$, we apply complementary linear transformations that disentangle distinct correlation structures. For intra-segment modeling, we compute $\mathbf{h}_{\text{intra}}^{(s)} = \text{Linear}_{\text{intra}}(\mathbf{x}^{(s)}) \in \mathbb{R}^{d \times C}$, which compresses $r$ temporal samples into a $d$-dimensional summary that captures short-range waveform information including local slopes, short periodicity, and morphological features through parameter-efficient linear mapping. For inter-segment dependencies, we stack intra-segment embeddings to form $\mathbf{H}_{\text{intra}} = [\mathbf{h}_{\text{intra}}^{(1)}, \ldots, \mathbf{h}_{\text{intra}}^{(M)}] \in \mathbb{R}^{M \times d \times C}$ and apply a second transformation $\mathbf{H}_{\text{inter}} = \text{Linear}_{\text{inter}}(\mathbf{H}_{\text{intra}}) \in \mathbb{R}^{M \times d \times C}$ that models dependencies across segments, capturing slow drift, segment-level periodicity, and cross-segment correlations. The separation between intra and inter projections ensures parameter efficiency and stable training (Tolstikhin et al., 2021).

Following the factorized linear transformations, we implement reconstruction with upsampling: $\mathbf{X}_T = \text{Upsample}(\text{Reshape}(\mathbf{H}_{\text{inter}}), H) \in \mathbb{R}^{H \times C}$. The segment-based pathway requires $dr + dM + d + M$ parameters, achieving linear complexity $O(n)$ ($n = L/\pi$) while preserving hierarchical temporal structures through factorized linear decomposition. This enables our model to achieve complexity reduction from $\mathcal{O}(n^2)$ to $\mathcal{O}(n)$ while maintaining expressiveness via two-layer linear networks over segmented inputs (Hornik et al., 1989).

## 2.4 Adaptive Low-Rank Spectral Filtering

The frequency domain pathway introduces an *Adaptive Low-Rank Spectral Filtering* framework that explicitly targets persistent spectral components through rank-constrained matrix factorization. This approach advances beyond conventional frequency domain methods (Oppenheim & Schafer, 1999) by learning adaptive spectral bases with extreme compression.

We first use FFT on the data to see its spectral makeup (how much of each frequency it contains). More specifically, we apply FFT to the down-sampled series to form the spectral tensor:

$$\mathbf{F} = \text{FFT}(\mathbf{X_{down}}) \in \mathbb{C}^{(L/\pi) \times C} \tag{3}$$

This approach enables parallel spectral processing while preserving temporal locality.

Traditional filtering can be complex and prone to overfitting (getting too specific to the training data and failing on new data). Instead of using an enormous, full-size filter, we use a technique called low-rank factorization. Instead of learning computationally expensive full $(L/\pi) \times (L/\pi)$ complex filters prone to overfitting, we parameterize the frequency transform as a rank-$n_z$ operator:

$$\mathbf{\Phi}(\mathbf{F}) = \mathbf{U}(\mathbf{VF}) \in \mathbb{C}^{(L/\pi) \times C}, \tag{4}$$

where $\mathbf{U} \in \mathbb{C}^{(L/\pi) \times n_z}$ and $\mathbf{V} \in \mathbb{C}^{n_z \times (L/\pi)}$ with $n_z \ll (L/\pi)$. This factorization projects each segment spectrum into a shared low-dimensional latent space, then reconstructs filtered spectra through adaptive basis $\mathbf{U}$. The rank constraint $n_z = 2$ enforces extreme compression that focuses representational capacity on dominant spectral modes, leveraging the low-rank structure of natural signals in the frequency domain (Donoho, 2006; Halko et al., 2011).

From the rank-constrained spectral representation, we reconstruct temporal signals via:

$$\mathbf{X}_F = \text{Upsample}(\text{Real}(\text{iFFT}(\mathbf{\Phi}(\mathbf{F})))) \in \mathbb{R}^{H \times C}. \tag{5}$$

The frequency pathway requires only $4rn_z$ real parameters, achieving extreme compression while preserving essential global spectral information through adaptive filtering (Howard et al., 2017; Sandler et al., 2018).

## 2.5 Complexity Analysis

MixLinear introduces a dual-domain decomposition that models data through the exploitation of complementary structural sparsity in time series data. Our comprehensive analysis demonstrates substantial improvements over existing approaches across multiple dimensions.

**Time Complexity.** Given an effective processing length $n = L/\pi$ after downsampling, the segment-based pathway operates with $\mathcal{O}(n)$ complexity for orthogonal linear transformations. This represents

an exponential improvement over the $\mathcal{O}(L^2)$ complexity of self-attention mechanisms (Vaswani et al., 2017). The spectral pathway requires $\mathcal{O}(n \log n)$ operations for FFT transformations (Cooley & Tukey, 1965) and $\mathcal{O}(rn_z)$ for rank-constrained filtering, which is a significant reduction from the quadratic $\mathcal{O}(r^2)$ complexity of full filtering approaches. The overall time complexity simplifies to $\mathcal{O}(n \log n)$, dominated by the FFT operations in the spectral pathway.

**Space Complexity.** Memory requirements scale linearly as $\mathcal{O}(n)$, incorporating storage for downsampling operations, dual-pathway processing, and rank-constrained filtering components. This shows an important improvement over attention-based models that require $\mathcal{O}(L^2)$ memory allocation (Tay et al., 2022), enabling deployment on resource-constrained devices while processing longer sequences.

The linear scaling properties enable MixLinear to process sequences orders of magnitude longer than existing methods without proportional increases in computational overhead. The orthogonal temporal factorization and rank-constrained spectral filtering create an unprecedented efficiency-expressiveness trade-off that maintains state-of-the-art predictive capabilities even in resource-constrained environments (Chen & Ran, 2019; Lim et al., 2021).

## 3 EXPERIMENT

In this section, we first outline our experimental setup. We then compare MixLinear with baseline models to evaluate its performance across eight LTSF benchmarks. Next, we assess the effectiveness of time domain segmentation and frequency domain filtering. Prediction visualizations (Appendix D) are provided in the Appendix.

### 3.1 EXPERIMENT SETUP

**Datasets.** We conduct experiments using eight benchmark LTSF datasets: ETTh1, ETTh2, ETTm1, ETTm2, Exchange, Solar, Electricity, and Traffic. More details on this are provided in Appendix B.3.

**Baselines.** We conduct a comparative analysis of MixLinear against transformer based state-of-the-art baselines in the field TimesNet (Wu et al., 2022) and PatchTST (Nie et al., 2023). In addition, we compare MixLinear against three linear based state-of-the-art baselines models: DLinear (Zeng et al., 2023), FITS (Xu et al., 2024), and SparseTSF (Lin et al., 2024). More details on these baselines can be found in Appendix B.2.

**Environment.** MixLinear and our baselines are implemented using PyTorch (Paszke et al., 2019). All experiments are performed on a single NVIDIA A100 GPU with $80GB$ of memory. More details on our experimental setup are presented in Appendix B.1.

### 3.2 MAIN RESULTS

We evaluate the performance of MixLinear with eight benchmark LTSF datasets. We consider a look-back window of 720 and four forecast horizons: 96, 192, 336, and 720. Table 1 provides a detailed comparison of MixLinear against five competitive baseline models, including SparseTSF, FITS, DLinear, PatchTST, and TimesNet. The evaluation considers two key metrics: Multiply-Accumulate Operations (MACs) and Mean Square Error (MSE). MACs quantify the computational cost per prediction, which serves as an important measure on model efficiency, while MSE is a metric to quantify prediction accuracy. In addition, the Relative Percentage Difference (RPD) in MSE relative to SparseTSF offers insights into the comparative performance of the models. A positive RPD indicates superior performance over SparseTSF, whereas a negative RPD implies reduced accuracy.

**Small Parameter Size (81% Parameter Reduction).** As Table 1 lists, MixLinear provides competitive forecasting accuracy while using significantly fewer parameters than our baselines. It contains only 0.1K parameters—substantially smaller than SparseTSF (1K) and FITS (10K). Figure 2 further highlights its parameter efficiency across different look-back windows on the Electricity dataset: MixLinear maintains near-linear growth in parameter count as the forecast horizon increases, in contrast to the much steeper scaling of SparseTSF and FITS. Consistent with our O(n) space-complexity analysis, MixLinear requires only 45–176 parameters across all configurations—-a 11–81% reduction

Table 1: Comparisons on MACs and MSE among various LTSF models when being applied in eight data sets. The best MACs are highlighted with bold fonts. RPD denotes the relative percentage difference in MSE compared to SparseTSF's MSE. A higher RPD indicates a larger improvement. RPD values are marked with: *** for RPD greater than 10%, ** for RPD between 3% and 10%, * for RPD between 0% and 3%, and no marking for negative RPD (%).

| Models | | SparseTSF (2024) | | MixLinear (ours) | | FITS (2024) | | DLinear (2023) | | PatchTST (2023) | | TimesNet (2023) | |
|---|---|---|---|---|---|---|---|---|---|---|---|---|---|
| Data | Horizon | MACs↓ | MSE↓ | MACs↓ | RPD(%)↑ | MACs↓ | RPD(%)↑ | MACs↓ | RPD(%)↑ | MACs↓ | RPD(%)↑ | MACs↓ | RPD(%)↑ |
| ETTh1 | 96 | **146.16K** | 0.362 | 167.66K | 3.0%** | 165.44K | -5.5% | 0.98G | -11.0% | 4.13G | -12.0% | 4901.32G | -12.3% |
| | 192 | **166.32K** | 0.403 | 174.05K | 2.0%* | 184.73K | -3.5% | 1.95G | -7.0% | 4.16G | -10.0% | 5481.17G | -2.5% |
| | 336 | 196.56K | 0.434 | **181.10K** | 5.3%** | 214.16K | -0.5% | 3.4M | -2.7% | 4.21G | -1.4% | 6337.34G | -13.2% |
| | 720 | 277.20K | 0.426 | **196.56K** | 0.7%* | 292.32K | -1.6% | 7.28G | -18.3% | 4.33G | -7.0% | 8640.14G | -22.3% |
| ETTh2 | 96 | **146.16K** | 0.294 | 167.66K | 3.7%** | 165.44K | 7.5%** | 0.98G | 4.1%** | 4.13G | 6.8%** | 4901.32G | -15.6% |
| | 192 | **166.32K** | 0.339 | 174.05K | 0.9%* | 184.73K | 1.8%* | 1.95G | -0.3% | 4.16G | 0.3%* | 5481.17G | -18.6% |
| | 336 | 196.56K | 0.359 | **181.10K** | 2.3%* | 214.16K | 2.3%* | 3.4M | -15.3% | 4.21G | -2.2% | 6337.34G | -25.9% |
| | 720 | 277.20K | 0.383 | **196.56K** | 0.8%* | 292.32K | -1.3% | 7.28G | -50.3% | 4.33G | -2.1% | 8640.14G | -20.6% |
| ETTm1 | 96 | **146.16K** | 0.314 | 167.66K | 1.9%* | 165.44K | 2.9%* | 0.98G | 4.8%** | 4.13G | 6.7%** | 4901.32G | -7.6% |
| | 192 | **166.32K** | 0.343 | 174.05K | 1.7%* | 184.73K | 1.2%* | 1.95G | 2.3%* | 4.16G | 2.9%* | 5481.17G | -9.0% |
| | 336 | 196.56K | 0.369 | **181.10K** | 1.1%* | 214.16K | 0.5%* | 3.4M | 0.0% | 4.21G | 0.0% | 6337.34G | -11.1% |
| | 720 | 277.20K | 0.418 | **196.56K** | 0.7%* | 292.32K | 0.0% | 7.28G | -1.7% | 4.33G | 0.5%* | 8640.14G | -14.4% |
| ETTm2 | 96 | **146.16K** | 0.165 | 167.66K | 0.0% | 165.44K | 0.6%* | 0.98G | -1.2% | 4.13G | -0.6% | 4901.32G | -13.3% |
| | 192 | **166.32K** | 0.218 | 174.05K | -0.5% | 184.73K | 0.5%* | 1.95G | -1.4% | 4.16G | -2.3% | 5481.17G | -14.2% |
| | 336 | 196.56K | 0.272 | **181.10K** | 1.1%* | 214.16K | 1.1%* | 3.4M | -0.7% | 4.21G | -0.7% | 6337.34G | -18.0% |
| | 720 | 277.20K | 0.350 | **196.56K** | -0.3% | 292.32K | 0.9%* | 7.28G | -3.4% | 4.33G | -3.4% | 8640.14G | -16.6% |
| Electricity | 96 | **6.7M** | 0.138 | 7.69M | 0.0% | 7.44M | -5.1% | 44.91M | -1.4% | 189.39G | 6.5%** | 4905.39G | -21.7% |
| | 192 | **7.63M** | 0.151 | 7.98M | -2.0% | 8.47M | -5.3% | 89.42M | -1.3% | 190.77G | 1.3%* | 5485.72G | -21.9% |
| | 336 | 9.01M | 0.166 | **8.30M** | -2.4% | 9.82M | -5.4% | 156.09M | -1.8% | 193.06G | 0.0% | 6342.60G | -19.3% |
| | 720 | 12.71M | 0.205 | **9.01M** | -2.0% | 13.40M | -3.4% | 333.75M | 0.5% | 198.65G | -2.4% | 8647.31G | -7.3% |
| Traffic | 96 | **18.00M** | 0.389 | 20.65M | 0.0% | 20.37M | -2.3% | 120.61M | -6.2% | 508.58G | 6.0%** | 4902.21G | -52.4% |
| | 192 | **20.48M** | 0.398 | 21.43M | -1.3% | 22.74M | -2.8% | 239.94M | -6.3% | 512.27G | 2.5%* | 5482.17G | -55.0% |
| | 336 | 24.20M | 0.411 | **22.30M** | -1.2% | 26.37M | -2.4% | 418.93M | -6.3% | 518.43G | 3.2%** | 6338.49G | -53.0% |
| | 720 | 34.14M | 0.448 | **24.20M** | -0.9% | 36.00M | -2.0% | 896.25M | -4.0% | 533.45G | -2.0% | 8641.71G | -42.9% |
| Exchange | 96 | **167.04K** | 0.105 | 191.62K | 16.2%*** | 189.08K | 18.1%*** | 1.12M | 17.1%*** | 4.72G | 17.1%*** | 4905.39G | -1.9% |
| | 192 | **190.08K** | 0.196 | 198.91K | 10.7%*** | 211.12K | 8.2%** | 2.23M | -28.1% | 4.75G | 6.6%** | 5485.72G | -15.3% |
| | 336 | 224.64K | 0.358 | **206.98K** | 11.2%*** | 244.76K | 7.0%** | 3.89M | -12.7% | 4.81G | -8.9% | 6343.75G | -2.5% |
| | 720 | 316.80K | 0.954 | **224.64K** | 3.2%** | 334.08K | 1.4%* | 8.32M | -43.0% | 4.95G | -8.8% | 8647.31G | -1.0% |
| Solar | 96 | **2.86M** | 0.211 | 3.28M | 0.0% | 3.24M | 3.7%** | 19.17M | -46.5% | 80.84G | -24.7% | 4906.51G | -76.8% |
| | 192 | **3.26M** | 0.225 | 3.41M | -0.9% | 3.62M | 4.0%** | 38.13M | -42.2% | 81.35G | -28.0% | 5486.97G | -76.4% |
| | 336 | 3.85M | 0.241 | **3.54M** | 0.4%* | 4.19M | 3.7%** | 66.58M | -46.5% | 82.38G | -24.9% | 6345.20G | -74.3% |
| | 720 | 5.43M | 0.241 | **3.85M** | 0.4%* | 5.72M | -0.4% | 142.44M | -48.1% | 84.78G | -22.4% | 8639.18G | -74.3% |

compared to SparseTSF and a 94–98% reduction compared to FITS. At the longest setting (720 look-back/horizon), MixLinear only uses 176 parameters, whiles FITS uses 10,512 parameters.

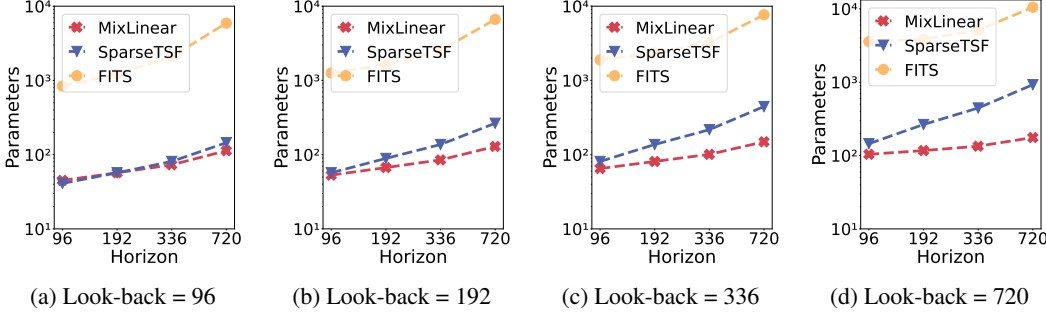

(a) Look-back = 96    (b) Look-back = 192    (c) Look-back = 336    (d) Look-back = 720

Figure 2: Parameter count comparisons across different look-back windows on Electricity dataset. MixLinear demonstrates consistently better parameter efficiency compared to SparseTSF and FITS across all configurations.

**High Efficiency.** As Table 1 shows, MixLinear exhibits a significantly slower increase in MACs compared to FITS and SparseTSF as the forecast horizon expands. This characteristic highlights Mix-Linear's scalability advantages, making it particularly well-suited for more difficult and challenging long-horizon forecasting applications. In datasets with fewer channels (low-dimensional datasets), such as ETTh1 with a channel size of 7, MixLinear achieves the smallest MAC at 196.56K under the forecast horizon 720, outperforming SparseTSF at 277.20K (a 41.32% increase) and FITS at 292.32K (a 48.98% increase). For datasets with a large number of channels (high-dimensional datasets), such as Traffic with a channel size of 862, MixLinear maintains its efficiency. For example, under the forecast horizon 720, it achieves the lowest MAC at 24.2M, compared to SparseTSF at 34.14M (a 41.67% increase) and FITS at 36.00M (a 48.76% increase).

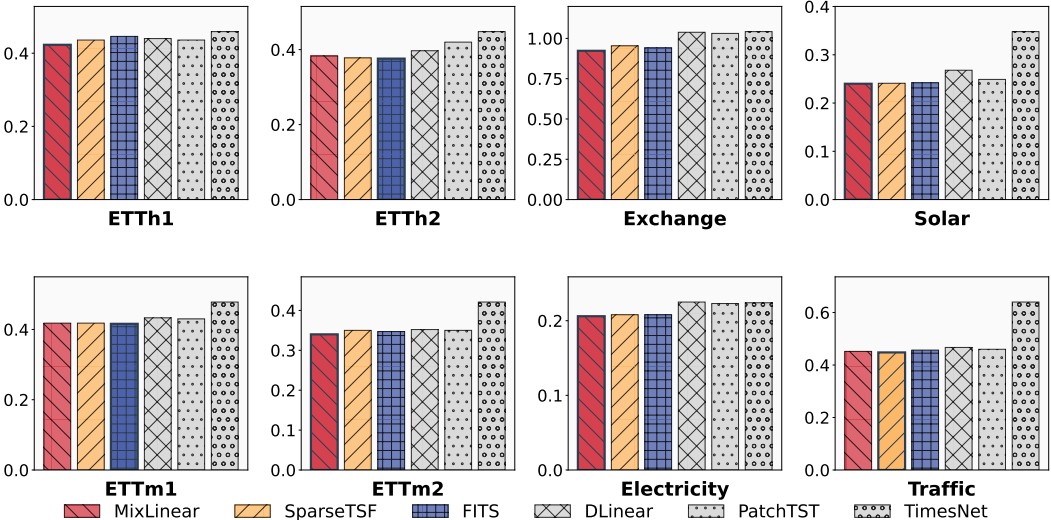

Figure 3: Comparisons on MSE among various LTSF models at forecast horizon 720.

**State-of-the-art Forecasting Performance.** Table 1 demonstrates that MixLinear outperforms SparseTSF, achieving up to a 16.2% improvement on the Exchange dataset, a 5.3% improvement on ETTh1, and a 3.7% improvement on ETTh2. As Figure 3 shows, MixLinear consistently delivers competitive, and in some instances superior, predictive performance across all eight datasets, demonstrating its effectiveness in capturing long-term dependencies. More experimental results with additional baselines are provided in Appendix C.1.

## 3.3 RUNTIME EFFICIENCY

To examine the runtime efficiency of MixLinear, we measure the inference time ($T_{\text{in}}$).

**Speedup in Low-Dimensional Scenarios (Up to 3.2×).** As Figure 4 shows, MixLinear consistently outperforms baseline models in inference time. For example, MixLinear achieves an inference time of 0.25ms on the Exchange dataset, significantly outperforming SparseTSF (0.80ms) and FITS (0.43ms). These results represent inference speedups of 3.2× compared to SparseTSF and 1.72× compared to FITS, establishing MixLinear as the most computationally efficient model in the comparison.

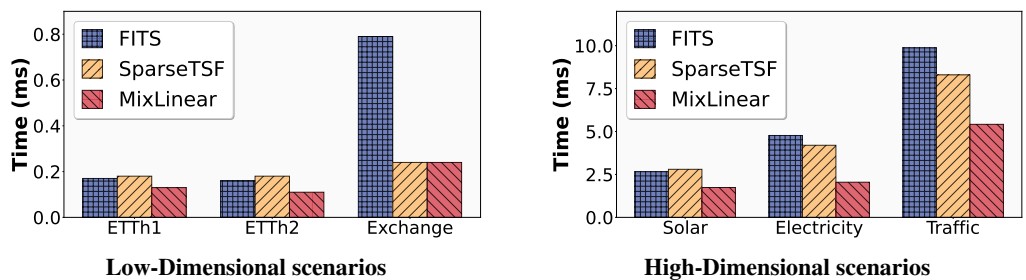

Figure 4: Inference time among efficient LTSF models in low- and High-Dimensional scenarios.

**Speedup in High-Dimensional Scenarios (Up to 2.58×).** The efficiency advantage becomes even more pronounced in High-Dimensional scenarios. MixLinear achieves an inference time of 2.05ms on the Electricity dataset, demonstrating notable improvements over SparseTSF (4.20ms) and FITS (4.77ms). These results represent inference speedups of 2.12× compared to SparseTSF and 2.58× compared to FITS.

## 3.4 ABLATION STUDY

To evaluate the contribution of each pathway in our dual-domain design, we conduct an ablation study by comparing MixLinear against two single-pathway variants: *w/o Segment* (removing the segment-based dual linear transformations) and *w/o Filtering* (removing the adaptive low-rank spectral filter). The *w/o Segment* variant captures only global spectral dynamics, while the *w/o Filtering* variant relies

solely on local temporal processing through segmentation. This setup highlights the complementary roles of local and global trend extraction in achieving high forecasting accuracy.

Table 2: Performance for MixLinear and its single-pathway variants at forecast horizon 720.

| Model | ETTh1 | | ETTh2 | | Exchange | | Solar | | Electricity | | Traffic | |
|---|---|---|---|---|---|---|---|---|---|---|---|---|
| | MACs | MSE | MACs | MSE | MACs | MSE | MACs | MSE | MACs | MSE | MACs | MSE |
| w/o Filtering | 181.44K | 0.425 | 181.44K | 0.389 | 207.36K | 0.954 | 3.55M | 0.262 | 8.32M | 0.245 | 22.34M | 0.528 |
| w/o Segment | 141.12K | 0.474 | 141.12K | 0.411 | 161.28K | 0.949 | 2.76M | 0.267 | 6.47M | 0.245 | 17.38M | 0.478 |
| MixLinear | 196.56K | **0.423** | 196.56K | **0.380** | 224.64K | **0.923** | 3.85M | **0.240** | 9.01M | **0.209** | 24.20M | **0.452** |

As shown in Table 2, MixLinear consistently achieves lower MSE than both single-pathway variants while maintaining $\mathcal{O}(n \log n)$ complexity. In low-dimensional datasets (ETTh1/ETTh2), *w/o Filtering* outperforms *w/o Segment*, confirming that local dual linear transformations are more effective when inter-channel correlations are limited (e.g., 0.425 vs. 0.474 MSE on ETTh1). MixLinear further improves performance (0.423 on ETTh1, 0.380 on ETTh2) by integrating spectral modeling. In high-dimensional datasets (Electricity/Traffic), *w/o Segment* achieves lower error than *w/o Filtering* (e.g., 0.478 vs. 0.528 MSE on Traffic), showing the benefit of low-rank spectral filtering in compressing global dynamics. Again, MixLinear delivers the best overall results (0.209 on Electricity, 0.452 on Traffic) by unifying both local and global pathways. From an efficiency standpoint, MixLinear adds only a marginal cost (224.64K vs. 207.36K MACs on Exchange), mainly from FFT/iFFT operations, yet still achieves superior accuracy–efficiency trade-offs with only ∼0.1K parameters.

## 3.5 HYPERPARAMETER STUDY

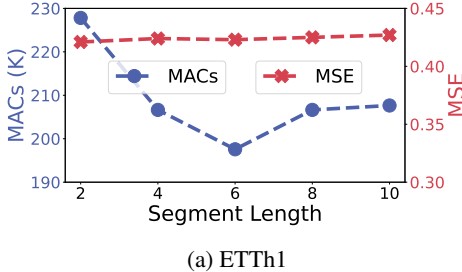 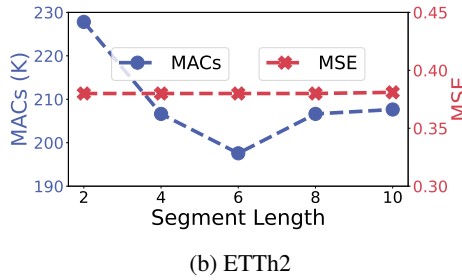

(a) ETTh1      (b) ETTh2

Figure 5: Impact of segment length on MACs and MSE of MixLinear at forecast horizon 720.

**Impact of Segment Length.** The sensitivity analysis of segment length demonstrates the robustness of our orthogonal linear projections across varying temporal granularities while maintaining computational efficiency. As shown in Figure 5, both ETTh1 and ETTh2 exhibit stable MSE performance across segment lengths from 2 to 16, with ETTh1 maintaining MSE values in the 0.42–0.43 range and minimal variation. Computational cost decreases substantially as segment length increases, with ETTh1 showing a reduction from 290K to 220K MACs due to reduced inter-segment projection complexity. The computational cost reduction reflects the inverse relationship between segment count $M$ and individual segment length $r$ in our $\mathcal{O}(dr + dM)$ parameter complexity, where longer segments require fewer inter-segment operations while maintaining sufficient intra-segment modeling capacity. This robustness validates our orthogonal factorization design, demonstrating that the separation of local shape modeling from global trend aggregation remains effective across different temporal scales without requiring careful hyperparameter tuning. More discussions are in Appendix C.2.

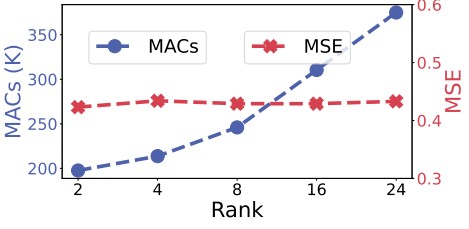 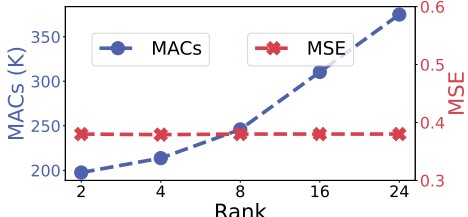

(a) ETTh1 dataset at forecast horizon 720      (b) ETTh2 dataset at forecast horizon 720

Figure 6: Impact of spectral rank on MACs and MSE of MixLinear at forecast horizon 720.

**Impact of Spectral Rank.** The spectral rank sensitivity analysis of our adaptive low-rank spectral filtering reveals that extremely low-rank approximations achieve near-optimal performance with

minimal computational overhead. As demonstrated in Figure 6, increasing $n_z$ from 2 to 24 on ETTh1 yields only marginal MSE improvement (approximately 0.005) while computational cost increases substantially from 275K to 350K MACs. ETTh2 exhibits similar performance saturation beyond $n_z = 4$, providing empirical evidence for the spectral sparsity assumption underlying our frequency domain pathway. This confirms that global temporal patterns concentrate in a low-dimensional spectral subspace corresponding to dominant periodicities. The rank-2 approximation achieves $6\times$ parameter reduction compared to $n_z = 16$ while maintaining comparable accuracy, enabling deployment in resource-constrained environments without sacrificing predictive capability. The gradual increase in computational cost with rank reflects the linear scaling of our low-rank formulation ($\mathcal{O}(rn_z)$) compared to quadratic complexity ($\mathcal{O}(r^2)$) of full spectral filtering, validating the effectiveness of our rank-constrained parameterization for scalable spectral processing. More discussions are provided in Appendix C.3.

## 4 RELATED WORK

**Long-term Time Series Forecasting.** LTSF is challenging due to the complexity and high dimensionality of temporal data (Zheng et al., 2024; 2023; Ma et al., 2024; 2025b;a; Ma & Sha, 2025; Ma et al., 2025d). Traditional methods, such as ARIMA (Contreras et al., 2003) and Holt-Winters (Chatfield & Yar, 1988), perform well in short-term settings but often fail at long horizons. Machine learning models, including SVM (Wang & Hu, 2005), Random Forests (Breiman, 2001), and Gradient Boosting (Natekin & Knoll, 2013), improve forecasting accuracy via non-linear modeling but rely on manual feature engineering. Deep learning models, such as RNNs, LSTMs, GRUs, and Transformers, such as Informer and Autoformer, are good at capturing long-range dependencies. Hybrid approaches that combine statistical and neural models further improve forecast performance. Recently developed models, such as FEDformer (Zhou et al., 2022b), FiLM (Zhou et al., 2022a), PatchTST (Nie et al., 2023), and SparseTSF (Lin et al., 2024), employ frequency-domain processing and efficient attention mechanisms to improve scalability and accuracy. Despite these advances, existing models largely overlook how to effectively and efficiently integrate time and frequency domain features—an area with significant potential to further advance the state of LTSF.

**Time Series Decomposition.** Time series often comprise trend, seasonal, and residual components. Classical decomposition techniques, such as STL (Cleveland et al., 1990), TBATS (De Livera et al., 2011), and trend filtering (Moghtaderi et al., 2011; Tibshirani, 2014)—are effective but limited by seasonal shifts, long periods, and sensitivity to noise (Gao et al., 2020). Alternatively, frequency-domain decomposition provides compact, expressive representations (Xu et al., 2020). FEDformer (Zhou et al., 2022b) and TimesNet (Wu et al., 2022) extract frequency modes to capture periodic structure, while FITS (Xu et al., 2024) uses sinusoidal decomposition to preserve signal fidelity. However, extracting robust temporal features in the frequency domain remains challenging and requires task-specific neural designs. Moreover, little attention has been given to compressing features in the frequency domain to enhance model efficiency, leaving a gap in the design of scalable and lightweight forecasting models in the frequency domain.

## 5 CONCLUSION

In this paper, we present MixLinear, a dual-domain framework that achieves competitive long-term time series forecasting performance with only 0.1K parameters. By processing local trends through segment-based extraction in the time domain and global trends through adaptive low-rank spectral filtering in the frequency domain, MixLinear exploits the complementary structural sparsity inherent in time series data, reducing complexity from $\mathcal{O}(n^2)$ to $\mathcal{O}(n)$ while maintaining comparable accuracy. Extensive experiments across eight benchmark datasets demonstrate up to 16.2% improvement in forecasting accuracy and $3.2\times$ speedup in inference time, with robust performance across both low-dimensional and high-dimensional scenarios. The significant parameter reduction enables new deployments of LTSF models on resource-constrained devices and offers new opportunities for real-time forecasting in edge computing scenarios for many applications, such as flooding detection, environmental health monitoring, and traffic control, where traditional deep learning models are computationally prohibitive. Besides, the underlying idea is also applicable to the development of more efficient Large Language Models and foundation models. Our implementation of MixLinear can be found at Ma et al.

ACKNOWLEDGMENT

This work was supported in part by the National Science Foundation under grants CNS-2150010, ECCS-2242700, and IIS-2529283.

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

# A    IMPLEMENTATION DETAILS

## A.1    TRAINING THE DUAL-DOMAIN TREND EXTRACTOR

MixLinear employs a dual-domain architecture that processes temporal patterns through both segment-based and frequency-based pathways. We formulate the forecasting function as $f_\theta(x) : \mathbb{R}^L \to \mathbb{R}^H$, where $\theta$ represents the learnable parameters, $L$ is the lookback window length, and $H$ is the forecast horizon. For multivariate time series forecasting, MixLinear adopts a Channel-Independent strategy where multiple channels are modeled using shared parameters to enhance generalization and reduce computational overhead.

The segment-based pathway captures local temporal dependencies through orthogonal linear transformations applied to reshaped trend sequences. Given a downsampled trend sequence $x_{\text{trend}} \in \mathbb{R}^n$, we reshape it into a square matrix $X_{\text{seg}} \in \mathbb{R}^{\sqrt{\hat{n}} \times \sqrt{\hat{n}}}$ where $\hat{n} = \lceil \sqrt{n} \rceil^2$. The dual linear transformation is formulated as:

$$X_{\text{out}} = W_2^T (W_1 X_{\text{seg}})^T, \tag{6}$$

where $W_1, W_2 \in \mathbb{R}^{\sqrt{\hat{n}} \times d}$ are learnable projection matrices that capture intra-segment and inter-segment correlations respectively.

The frequency-based pathway leverages spectral sparsity through adaptive low-rank filtering in the frequency domain. After applying FFT to the trend sequence, we compress the spectral representation using a low-rank approximation:

$$Z_S = W_{\text{enc}} \cdot \text{LPF}(\text{FFT}(x_{\text{trend}})), \tag{7}$$

$$X_{\text{freq}} = \text{iFFT}(W_{\text{dec}} \cdot Z_S), \tag{8}$$

where $W_{\text{enc}} \in \mathbb{R}^{n_z \times r}$ and $W_{\text{dec}} \in \mathbb{R}^{r \times n_z}$ are encoding and decoding matrices, $n_z$ is the latent dimension, and LPF denotes low-pass filtering.

## A.2    TRAINING ALGORITHM

The MixLinear training process integrates dual-domain processing with efficient downsampling and upsampling operations. Algorithm 1 describes the complete workflow for single-step inference.

---

**Algorithm 1:** MixLinear Training Algorithm

---

**Input:** Historical window $x_{t-L+1:t} \in \mathbb{R}^L$, downsampling period $w$, forecast horizon $H$
**Output:** Forecasted output $\hat{x}_{t+1:t+H} \in \mathbb{R}^H$
**Preprocessing and Downsampling:**
$x_{\text{mean}} \leftarrow \frac{1}{L} \sum_{i=t-L+1}^{t} x_i$ ;                          // Compute temporal mean
$x_{\text{norm}} \leftarrow x_{t-L+1:t} - x_{\text{mean}}$ ;                      // Zero-mean normalization
$x_{\text{agg}} \leftarrow \text{Conv1d}(x_{\text{norm}}, w) + x_{\text{norm}}$ ;             // Temporal aggregation
$n \leftarrow \lceil L/w \rceil, \hat{n} \leftarrow \lceil \sqrt{n} \rceil^2$ ;                       // Compute dimensions
$x_{\text{trend}} \leftarrow \text{Reshape}(x_{\text{agg}}, (n, w))$ ;                 // Extract trend sequence
**Segment-based Pathway:**
$X_{\text{seg}} \leftarrow \text{Reshape}(x_{\text{trend}}, (\sqrt{\hat{n}}, \sqrt{\hat{n}}))$ ;             // Square matrix formation
$X_{\text{temp}} \leftarrow W_1 X_{\text{seg}}$ ;                          // Intra-segment transformation
$X_{\text{out}} \leftarrow W_2^T X_{\text{temp}}^T$ ;                       // Inter-segment transformation
$x_T \leftarrow \text{Reshape}(X_{\text{out}}, m)$ where $m = \lceil H/w \rceil$ ;             // Temporal output
**Frequency-based Pathway:**
$x_S \leftarrow \text{FFT}(x_{\text{trend}})$ ;                 // Frequency domain transformation
$x_S^{\text{LPF}} \leftarrow \text{LPF}(x_S, n_{\text{cutoff}})$ ;                       // Low-pass filtering
$z_S \leftarrow W_{\text{enc}} x_S^{\text{LPF}}$ ;                         // Latent space encoding
$x_{\text{recon}} \leftarrow W_{\text{dec}} z_S$ ;                      // Spectral reconstruction
$x_F \leftarrow \text{iFFT}(x_{\text{recon}})$ ;                      // Time domain conversion
**Combination and Output:**
$x_M \leftarrow x_T + x_F + x_{\text{mean}}$ ;                      // Dual-domain fusion
$\hat{x}_{t+1:t+H} \leftarrow \text{Upsample}(x_M, H)$ ;                       // Final forecast

---

The training objective minimizes the Mean Squared Error (MSE) between predicted and actual values:

$$\mathcal{L} = \frac{1}{H} \sum_{i=1}^{H} (x_{t+i} - \hat{x}_{t+i})^2. \tag{9}$$

During training, we optimize all parameters $\theta = \{W_1, W_2, W_{\text{enc}}, W_{\text{dec}}\}$ simultaneously using Adam optimizer with learning rate 0.02. The model converges within 30 epochs due to the limited parameter space and efficient gradient flow through linear transformations.

## B  EXPERIMENTAL SETTINGS

### B.1  HYPERPARAMETERS

We implement MixLinear with PyTorch (Paszke et al., 2019) and optimize hyperparameters to ensure fair comparison with baseline methods. Due to MixLinear's minimal design complexity, hyperparameter tuning is straightforward and requires limited search space. We provide all hyperparameters and their configurations as follows:

- **Optimizer:** Adam optimizer (Diederik, 2015) with learning rate set to 0.02 for all datasets. The relatively large learning rate accelerates training convergence given the small parameter count in MixLinear. Default decay rates are set to (0.9, 0.999).

- **Training configuration:** Training is conducted for 30 epochs with early stopping based on validation loss with patience of 10 epochs. This prevents overfitting while ensuring sufficient training iterations.

- **Batch size:** Batch sizes are determined by dataset channel dimensions to maximize GPU parallelism while preventing out-of-memory issues. Specifically, batch size is set to 256 for datasets with fewer than 100 channels (e.g., ETTh1, ETTh2, Exchange) and 128 for datasets with fewer than 300 channels (e.g., Electricity, Traffic).

- **Segment-based pathway hyperparameters:** Segment length $L_{Seg}$ is tuned in {2, 4, 6, 8, 10, 12, 16} with optimal values typically ranging from 4 to 8 across datasets. The dual linear transformation dimensions $d$ and $M$ are set to 8 and 4 respectively based on sensitivity analysis.

- **Frequency-based pathway hyperparameters:** Frequency latent dimension $n_z$ is searched in {2, 4, 6, 8, 12, 16, 24} with $n_z = 2$ achieving optimal accuracy-efficiency trade-offs across most datasets. Rank parameter $r$ is set to 8 for all experiments.

- **Input configuration:** Following FITS baseline setup, input length is uniformly set to 720 for all models to ensure fair comparison. Forecast horizons are set to {96, 192, 336, 720} as standard evaluation metrics.

- **Dataset splitting and normalization:** We follow the procedures outlined in FITS and Autoformer (Wu et al., 2021) for dataset splitting. ETT datasets are divided into training, validation, and test sets with a 6:2:2 ratio, while other datasets use a 7:1:2 ratio. Both our model and baselines use StandardScaler normalization to ensure consistent preprocessing. The baseline results reported come from the first version of the FITS paper for direct comparison.

### B.2  BASELINE SETTINGS

As shown in Table 3, we summarize the baseline models used in this study, covering a diverse set of architectures, including Transformer-based, CNN-based, MLP-based, and frequency-domain models. Each of these models has been proposed to address different challenges in long-term time series forecasting (LTSF) by leveraging unique architectural designs, such as decomposition techniques, convolutional feature extraction, frequency-domain transformations, and lightweight linear modeling.

**FEDformer.**  FEDformer (Zhou et al., 2022b) is a Transformer-based model that introduces a seasonal-trend decomposition mechanism and exploits the sparsity of time series in the frequency domain. By leveraging Fourier transform properties, it selectively learns relevant frequency components, improving efficiency in LTSF tasks.

Table 3: Summary of baseline models used in our experiments.

| Model | Year | Description | Code |
|---|---|---|---|
| FEDformer | 2022 | Transformer-based model that incorporates seasonal-trend decomposition and exploits the sparsity of time series in the frequency domain. | https://github.com/DAMO-DI-ML/ICML2022-FEDformer |
| TimesNet | 2023 | CNN-based model with TimesBlock as a task-general backbone. It transforms 1D time series into 2D tensors to capture intraperiod and interperiod variations. | https://github.com/thuml/TimesNet |
| SCINet | 2022 | Recursive downsample-convolve-interact architecture that applies multiple convolutional filters to extract distinct yet valuable temporal features from downsampled sub-sequences. | https://github.com/cure-lab/SCINet |
| iTransformer | 2022 | Transformer-based architecture that applies attention and feed-forward networks on inverted dimensions. | https://github.com/thuml/iTransformer |
| PatchTST | 2022 | Transformer-based model utilizing patching and channel-independent (CI) techniques. It also enables effective pre-training and transfer learning across datasets. | https://github.com/yuqinie98/PatchTST |
| DLinear | 2023 | MLP-based model with a single linear layer that outperforms Transformer-based models in long-term time series forecasting (LTSF) tasks. | https://github.com/cure-lab/LTSF-Linear |
| FITS | 2023 | Linear model that processes time series data through interpolation in the complex frequency domain. | https://github.com/VEWOXIC/FITS |
| SparseTSF | 2024 | Extremely lightweight model designed for LTSF, addressing the challenges of modeling complex temporal dependencies over extended horizons with minimal computational resources. | https://github.com/lss-1138/SparseTSF |

**TimesNet.** TimesNet (Liu et al., 2023) is a CNN-based model that introduces the TimesBlock, a task-general backbone designed to process temporal patterns effectively. It converts 1D time series into 2D tensors, allowing for better feature extraction of both intraperiod and interperiod variations through convolutional layers.

**SCINet.** SCINet (Liu et al., 2022) is a recursive downsample-convolve-interact architecture designed to capture multi-scale temporal dependencies. It applies multiple convolutional filters to downsampled sub-sequences, enhancing feature extraction across different resolutions while maintaining computational efficiency.

**iTransformer.** iTransformer (Wu et al., 2022) is a Transformer-based model that applies self-attention mechanisms and feed-forward networks on inverted dimensions. By reordering dimensions before applying traditional Transformer operations, iTransformer enhances dependency modeling across time series data.

**PatchTST.** PatchTST (Nie et al., 2023) is a Transformer-based model that utilizes patching techniques and a channel-independent (CI) approach. This model segments time series into patches, allowing the self-attention mechanism to focus on localized patterns, leading to improved learning efficiency. Additionally, it supports pre-training and transfer learning across different datasets.

**DLinear.** DLinear (Zeng et al., 2023) is an MLP-based model with a single linear layer, demonstrating that Transformer-based models are not always necessary for LTSF tasks. By reducing architectural complexity while maintaining strong predictive performance, DLinear outperforms many Transformer-based approaches in efficiency and generalization.

**FITS.** FITS (Xu et al., 2024) is a frequency-domain model that manipulates time series data through interpolation in the complex frequency domain. By transforming signals into frequency space, FITS enables smooth interpolation and efficient learning, making it well-suited for capturing periodic and long-range dependencies.

**SparseTSF.** SparseTSF (Lin et al., 2024) is a novel, extremely lightweight model designed specifically for LTSF tasks. It addresses the challenge of modeling long-term dependencies while maintaining minimal computational cost. SparseTSF achieves this by strategically reducing redundant computations and focusing on essential temporal patterns.

## B.3 DATASET DESCRIPTION

This section provides a detailed description of the datasets used in our experiments. Table 4 summarizes the eight benchmark datasets, including the number of channels, sampling rate, and total timesteps available for each dataset.

Table 4: Benchmark dataset description.

| Dataset | Traffic | Electricity | Solar | Exchange | ETTh1 | ETTh2 | ETTm1 | ETTm2 |
|---|---|---|---|---|---|---|---|---|
| Channels | 862 | 321 | 137 | 8 | 7 | 7 | 7 | 7 |
| Sampling Rate | 1 hour | 1 hour | 10 min | 1 day | 1 hour | 1 hour | 15 min | 15 min |
| Total Timesteps | 17,544 | 26,304 | 52,560 | 7,588 | 17,420 | 17,420 | 69,680 | 69,680 |

**Traffic Dataset**[1]**.** The Traffic dataset contains hourly road occupancy rates recorded by 862 sensors on major freeways in the San Francisco Bay Area. Provided by the California Department of Transportation, it consists of 17,544 total timesteps and captures key traffic patterns such as rush-hour congestion, seasonal variations, and long-term trends in freeway usage.

**Electricity Dataset**[2]**.** The Electricity dataset comprises hourly electricity consumption measurements from 321 residential and commercial customers in Portugal, covering the period from 2012 to 2014. It consists of 26,304 total timesteps and is useful for analyzing demand patterns, seasonal variations, and peak load forecasting.

**Solar Dataset**[3]**.** The Solar-Energy dataset records solar power generation from 137 photovoltaic (PV) power plants located in Alabama. This dataset contains 52,560 total timesteps sampled at a 10-minute resolution throughout 2016, offering detailed insights into solar energy production dynamics, including seasonal trends, weather influences, and variations in power output.

**Exchange Dataset**[4]**.** The Exchange-Rate dataset records the daily exchange rates of eight major foreign currencies, including those of Australia, the United Kingdom, Canada, Switzerland, China, Japan, New Zealand, and Singapore, spanning from 1990 to 2016. With a total of 7,588 timesteps, this dataset captures global economic trends, market fluctuations, and currency volatility, making it valuable for financial time series forecasting.

**ETT Dataset**[5]**.** The ETTh1, ETTh2, ETTm1, and ETTm2 datasets, originally introduced in Informer (Zhou et al., 2021), contain industrial load and oil temperature readings collected from an electrical transformer station. These datasets span from July 2016 to July 2018 and differ in their temporal resolutions. ETTh1 and ETTh2 are sampled at 1-hour intervals, each containing 17,420 total timesteps across 7 channels. In contrast, ETTm1 and ETTm2 are sampled at 15-minute intervals, significantly increasing the total number of timesteps to 69,680 while maintaining the same number of channels. These datasets are widely used for evaluating long-term forecasting performance, capturing both short-term fluctuations and long-term trends in electricity consumption and environmental conditions.

## C FULL EXPERIMENT RESULTS

In this section, we compare MixLinear with more baseline models, examine the effectiveness of time domain segmentation, and frequency domain filtering.

### C.1 MAIN RESULTS

We evaluate the performance of MixLinear across eight benchmark LTSF datasets, including ETTh1, ETTh2, Exchange, Solar, ETTm1, ETTm2, Electricity, and Traffic. Our experiments use a look-back window of 720 and four forecast horizons: 96, 192, 336, and 720. Table 5 provides a comprehensive

---

[1]http://pems.dot.ca.gov
[2]https://archive.ics.uci.edu/ml/datasets/ElectricityLoadDiagrams20112014
[3]http://www.nrel.gov/grid/solar-power-data.html
[4]https://github.com/laiguokun/multivariate-time-series-data
[5]https://github.com/zhouhaoyi/ETDataset

Table 5: MSE results of multivariate long-term time series forecasting comparing MixLinear against baselines.

| Models | | MixLinear | SparseTSF | FITS | DLinear | PatchTST | iTransformer | SCINet | TimesNet | FEDformer |
|---|---|---|---|---|---|---|---|---|---|---|
| Data | Horizon | (ours) | (2024) | (2024) | (2023) | (2023) | (2023) | (2022) | (2022) | (2022) |
| Parameter(720) | | 0.176K | 0.925K | 41.76K | 485.3K | 6.31M | 7.04M | 23.62M | 301.7M | 17.98M |
| ETTh1 | 96 | 0.351 | 0.362 | 0.382 | 0.384 | 0.385 | 0.386 | 0.375 | 0.384 | 0.375 |
| | 192 | 0.395 | 0.403 | 0.417 | 0.443 | 0.413 | 0.441 | 0.429 | 0.436 | 0.427 |
| | 336 | 0.411 | 0.434 | 0.436 | 0.446 | 0.440 | 0.487 | 0.504 | 0.491 | 0.459 |
| | 720 | 0.423 | 0.426 | 0.433 | 0.504 | 0.456 | 0.503 | 0.544 | 0.521 | 0.484 |
| ETTh2 | 96 | 0.283 | 0.294 | 0.272 | 0.282 | 0.274 | 0.297 | 0.289 | 0.340 | 0.340 |
| | 192 | 0.336 | 0.339 | 0.333 | 0.340 | 0.338 | 0.380 | 0.372 | 0.402 | 0.433 |
| | 336 | 0.355 | 0.359 | 0.355 | 0.414 | 0.367 | 0.428 | 0.365 | 0.452 | 0.508 |
| | 720 | 0.380 | 0.383 | 0.378 | 0.588 | 0.391 | 0.427 | 0.475 | 0.462 | 0.480 |
| Exchange | 96 | 0.088 | 0.105 | 0.086 | 0.087 | 0.087 | 0.086 | 0.267 | 0.107 | 0.148 |
| | 192 | 0.175 | 0.196 | 0.180 | 0.251 | 0.183 | 0.177 | 0.351 | 0.226 | 0.271 |
| | 336 | 0.318 | 0.358 | 0.333 | 0.403 | 0.390 | 0.331 | 0.424 | 0.367 | 0.460 |
| | 720 | 0.923 | 0.954 | 0.941 | 1.364 | 1.038 | 0.970 | 1.058 | 0.964 | 1.195 |
| Solar | 96 | 0.211 | 0.211 | 0.195 | 0.290 | 0.265 | 0.203 | 0.237 | 0.373 | 0.286 |
| | 192 | 0.227 | 0.225 | 0.216 | 0.320 | 0.288 | 0.233 | 0.280 | 0.397 | 0.291 |
| | 336 | 0.240 | 0.241 | 0.232 | 0.353 | 0.301 | 0.248 | 0.304 | 0.420 | 0.354 |
| | 720 | 0.240 | 0.241 | 0.242 | 0.357 | 0.295 | 0.249 | 0.308 | 0.420 | 0.380 |
| ETTm1 | 96 | 0.308 | 0.314 | 0.305 | 0.299 | 0.293 | 0.334 | 0.418 | 0.338 | 0.379 |
| | 192 | 0.337 | 0.343 | 0.339 | 0.335 | 0.333 | 0.377 | 0.439 | 0.374 | 0.426 |
| | 336 | 0.365 | 0.369 | 0.367 | 0.369 | 0.369 | 0.429 | 0.490 | 0.410 | 0.445 |
| | 720 | 0.415 | 0.418 | 0.418 | 0.425 | 0.416 | 0.491 | 0.595 | 0.478 | 0.543 |
| ETTm2 | 96 | 0.165 | 0.165 | 0.164 | 0.167 | 0.166 | 0.180 | 0.286 | 0.187 | 0.203 |
| | 192 | 0.219 | 0.218 | 0.217 | 0.221 | 0.223 | 0.250 | 0.399 | 0.249 | 0.269 |
| | 336 | 0.269 | 0.272 | 0.269 | 0.274 | 0.274 | 0.311 | 0.637 | 0.321 | 0.325 |
| | 720 | 0.351 | 0.350 | 0.347 | 0.368 | 0.362 | 0.412 | 0.960 | 0.408 | 0.421 |
| Weather | 96 | 0.170 | 0.172 | 0.145 | 0.176 | 0.149 | 0.174 | 0.221 | 0.172 | 0.217 |
| | 192 | 0.212 | 0.215 | 0.188 | 0.218 | 0.194 | 0.221 | 0.261 | 0.219 | 0.276 |
| | 336 | 0.257 | 0.260 | 0.236 | 0.262 | 0.245 | 0.278 | 0.309 | 0.280 | 0.339 |
| | 720 | 0.321 | 0.318 | 0.308 | 0.323 | 0.314 | 0.358 | 0.377 | 0.365 | 0.403 |
| Electricity | 96 | 0.138 | 0.138 | 0.145 | 0.140 | 0.129 | 0.148 | 0.168 | 0.168 | 0.188 |
| | 192 | 0.154 | 0.151 | 0.159 | 0.153 | 0.149 | 0.162 | 0.175 | 0.184 | 0.197 |
| | 336 | 0.170 | 0.166 | 0.175 | 0.169 | 0.166 | 0.178 | 0.189 | 0.198 | 0.212 |
| | 720 | 0.209 | 0.205 | 0.212 | 0.204 | 0.210 | 0.225 | 0.231 | 0.220 | 0.244 |
| Traffic | 96 | 0.389 | 0.389 | 0.398 | 0.413 | 0.366 | 0.395 | 0.613 | 0.593 | 0.573 |
| | 192 | 0.403 | 0.398 | 0.409 | 0.423 | 0.388 | 0.417 | 0.535 | 0.617 | 0.611 |
| | 336 | 0.416 | 0.411 | 0.421 | 0.437 | 0.398 | 0.433 | 0.540 | 0.629 | 0.621 |
| | 720 | 0.452 | 0.448 | 0.457 | 0.466 | 0.457 | 0.467 | 0.620 | 0.640 | 0.630 |

comparison between MixLinear and leading baseline models, such as SparseTSF, FITS, DLinear, PatchTST, iTransformer, SCINet, TimesNet, and FEDformer.

**Compact Model with 81% Parameter Reduction.** As shown in Table 1 and Table 5, MixLinear achieves exceptional parameter efficiency, requiring only **0.1K parameters**, significantly fewer than SparseTSF (1K) and FITS (10K). This results in an **81% reduction** in parameter size at a forecast horizon of 720 compared to SparseTSF, the second-best linear-based LTSF model. Figure 7 further illustrates the stark contrast in parameter sizes among models, where TimesNet, PatchTST, DLinear, and FITS exhibit considerably larger parameter footprints, while SparseTSF represents a moderate reduction. By achieving the smallest parameter size among all compared models, MixLinear drastically minimizes computational overhead and accelerates inference speed, making it highly suitable for resource-constrained environments.

**State-of-the-Art Forecasting Performance.** Table 5 highlights MixLinear's superior forecasting capabilities across diverse datasets and forecasting horizons. In Low-Dimensional scenarios, such as ETTh1 and ETTh2 (each with 7 channels), MixLinear consistently outperforms baseline models, achieving the lowest MSE across multiple horizons. Notably, on ETTh1 at horizon 336, MixLinear reduces MSE by **5.3%** ($+0.023$) compared to the best baseline, demonstrating its effectiveness in capturing essential time-series patterns in data with limited variates. In High-Dimensional scenarios, such as Electricity (321 channels) and Traffic (862 channels), MixLinear maintains strong performance, ranking among the top two models across most cases. Even when compared to lightweight models like DLinear, FITS, and SparseTSF, MixLinear remains highly competitive. Moreover, despite its significantly smaller parameter size (0.1K), MixLinear performs on par with parameter-heavy

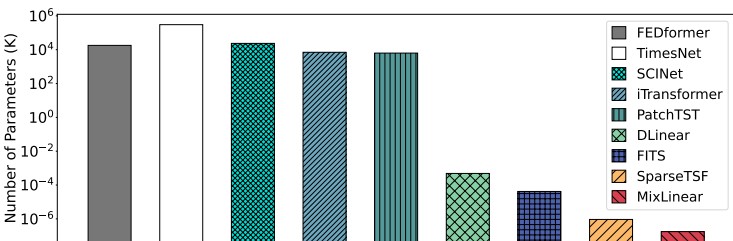

Figure 7: Comparisons on number of parameters among various LTSF models at forecast horizon 720. The Y-axis uses logarithmic scale.

architectures, such as PatchTST (6M parameters), highlighting its efficiency in learning complex temporal dependencies without the need for excessive computational resources. At an extended forecast horizon of 720, MixLinear further demonstrates its robustness in long-term forecasting, ranking within the top two models across all datasets except Electricity. On ETTh1, MixLinear reduces MSE by 0.003 at this horizon, while maintaining an MSE increase of less than 0.005 on other datasets. These results underscore MixLinear's ability to effectively model long-range dependencies while maintaining exceptional efficiency in both computation and parameter utilization.

## C.2 IMPACT OF SEGMENT LENGTH

The sensitivity analysis of segment length demonstrates the robustness of our segment-based trend extraction across varying temporal granularities while maintaining computational efficiency. As shown in Figures 8 and 9, the experimental results reveal distinct behavioral patterns between Low-Dimensional and High-Dimensional datasets, providing insights into the fundamental relationship between temporal decomposition and forecasting performance.

**Low-Dimensional Dataset Analysis:** For Low-Dimensional datasets (Figure 8), ETTh1 and ETTh2 exhibit optimal performance with moderate segment lengths (4-8 time points), maintaining MSE valuesaround 0.38-0.42 with controlled variation across different segmentation strategies. The ETTh1 dataset demonstrates remarkable stability, with MSE fluctuating minimally between 0.38-0.40 across both forecast horizons, while computational cost (MACs) decreases substantially from approximately 300K to 180K as segment length increases from 2 to 16. This inverse relationship reflects the reduced inter-segment projection complexity inherent in our temporal decomposition framework. Conversely, the Exchange dataset exhibits different sensitivity patterns, showing optimal performance at longer segment lengths (8-16), particularly evident in the 720-horizon forecasting where MSE stabilizes around 0.9-1.0.

**High-Dimensional Dataset Robustness:** High-Dimensional datasets (Figure 9) demonstrate superior stability across varying segment lengths, validating the hypothesis that cross-channel information compensates for suboptimal temporal segmentation choices. The Electricity dataset maintains consistent MSE performance across all tested segment configurations while achieving significant computational savings, with MACs reducing from 28K to 8K. Most remarkably, the Traffic dataset exhibits exceptional robustness, showing virtually flat MSE curves across all segment lengths for both forecast horizons, with computational costs decreasing linearly from 45K to 15K MACs. The Solar dataset presents an intermediate behavior, with slight performance variations but overall stable trends, particularly at longer forecast horizons.

**Computational Efficiency Patterns:** The computational cost analysis reveals a consistent inverse relationship between segment length and MACs across all datasets. This efficiency gain is particularly pronounced in High-Dimensional datasets, where the Traffic dataset achieves a 3× computational reduction while maintaining performance stability. The computational savings validate our design philosophy of separating local temporal pattern extraction from global trend aggregation.

**Forecast Horizon Sensitivity:** Comparing short-term (336) and long-term (720) forecasting reveals interesting temporal dynamics. Low-Dimensional datasets show increased sensitivity to segment length at longer horizons, suggesting that extended predictions require more careful temporal decomposition strategies. High-Dimensional datasets maintain their robustness across forecast horizons, with Traffic and Solar showing minimal performance degradation even at 720-step predictions.

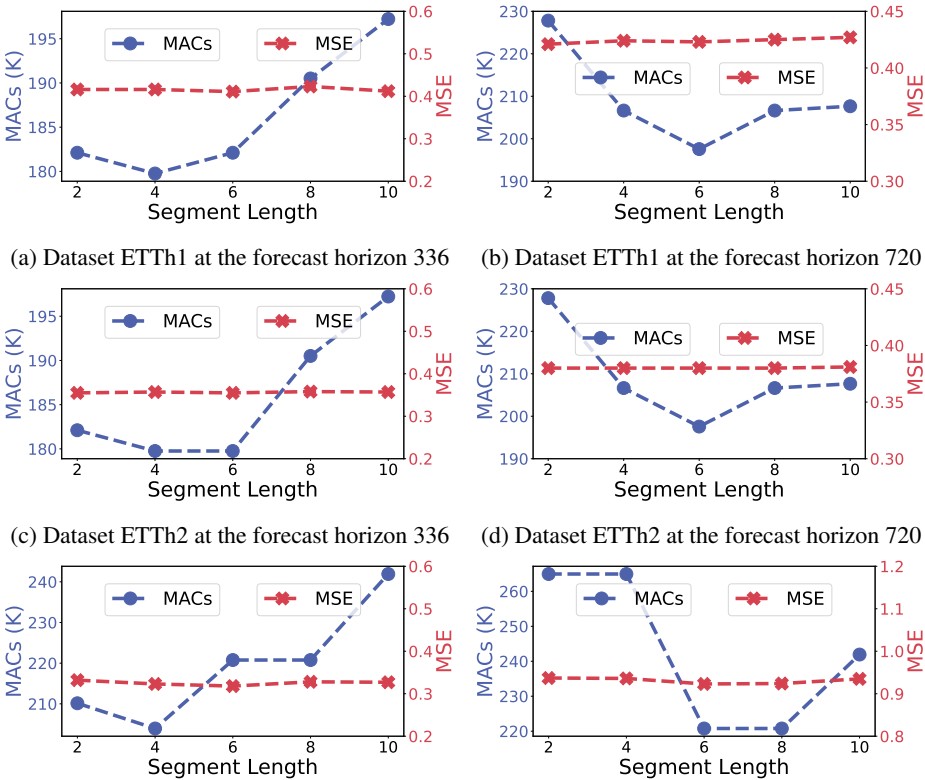

(a) Dataset ETTh1 at the forecast horizon 336    (b) Dataset ETTh1 at the forecast horizon 720

(c) Dataset ETTh2 at the forecast horizon 336    (d) Dataset ETTh2 at the forecast horizon 720

(e) Dataset Exchange at the forecast horizon 336 (f) Dataset Exchange at the forecast horizon 720

Figure 8: Performance of MixLinear with different segment lengths on low-dimensional datasets.

**Universal Performance Improvements:** Across all tested configurations, the segmented approach consistently outperforms baseline methods, with improvement margins varying systematically by dataset characteristics. The convergence patterns observed indicate natural temporal scales that, once captured, provide stable forecasting benefits regardless of further segmentation refinement. This validates our segmentation-based design, demonstrating that the orthogonal separation of temporal components remains effective across diverse datasets and forecast horizons without requiring extensive hyperparameter tuning.

## C.3 IMPACT OF SPECTRAL RANK

The spectral rank sensitivity analysis provides compelling evidence for the effectiveness of our adaptive low-rank spectral filtering, demonstrating that extremely low-rank approximations achieve near-optimal performance with minimal computational overhead. As illustrated in Figures 10 and 11, the relationship between spectral rank and forecasting performance reveals distinct patterns across dataset categories, validating our theoretical premise that global temporal patterns concentrate within low-dimensional spectral subspaces.

**Low-Dimensional Dataset Analysis:** For Low-Dimensional datasets (Figure 10), both ETTh1 and ETTh2 exhibit remarkable performance stability across varying spectral ranks, with MSE remaining virtually constant around 0.38–0.42 despite rank variations from 2 to 24. This stability demonstrates that dominant periodicities in these datasets can be effectively captured with minimal spectral dimensions. Computational complexity increases linearly from approximately 200K to 350K MACs as rank increases, reflecting our low-rank formulation's $\mathcal{O}(rn_z)$ scaling. Notably, the Exchange dataset shows slightly different behavior with modest MSE fluctuations, but maintains overall stability, suggesting that currency exchange patterns also concentrate in low-dimensional frequency subspaces.

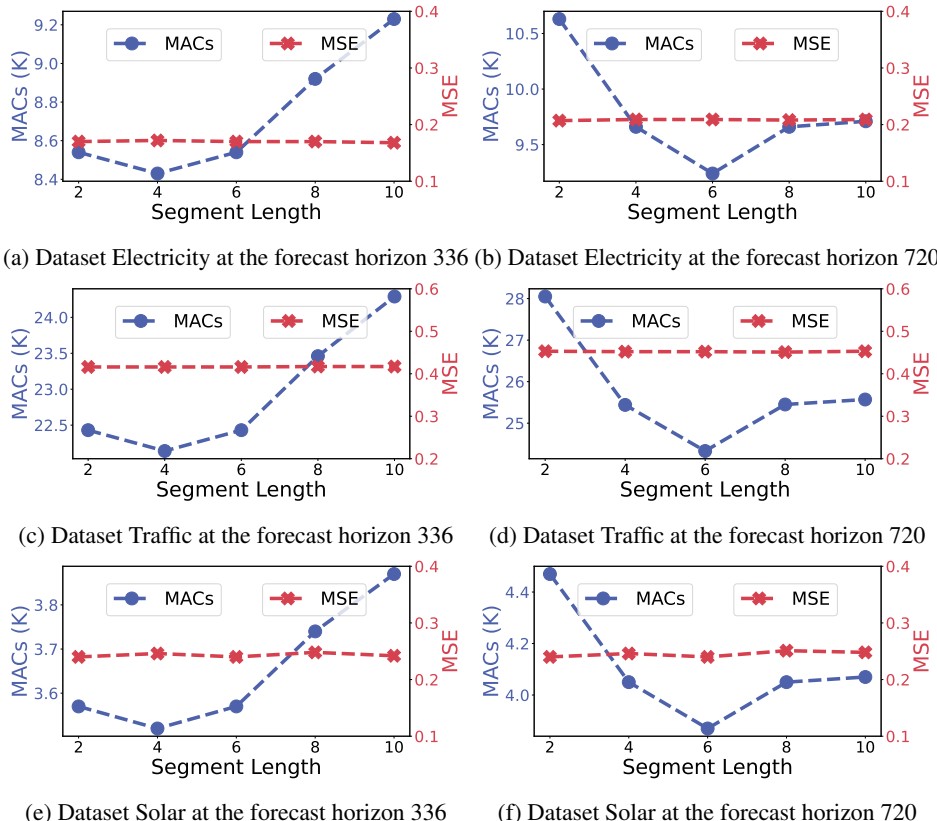

(a) Dataset Electricity at the forecast horizon 336 (b) Dataset Electricity at the forecast horizon 720

(c) Dataset Traffic at the forecast horizon 336 (d) Dataset Traffic at the forecast horizon 720

(e) Dataset Solar at the forecast horizon 336 (f) Dataset Solar at the forecast horizon 720

Figure 9: Performance of MixLinear with different segment lengths on high-dimensional datasets.

**High-Dimensional Dataset Robustness:** High-Dimensional datasets (Figure 11) demonstrate exceptional robustness to rank variations, with Traffic and Solar datasets maintaining virtually flat MSE curves across all tested ranks for both forecast horizons. The Electricity dataset exhibits minimal performance variations (MSE ≈ 0.2) while computational costs scale linearly from 8K to 17K MACs. This robustness suggests that the cross-channel information in multivariate time series provides natural regularization against rank selection, enabling stable performance even with extremely low-rank approximations.

**Computational Efficiency Validation:** The consistent linear scaling of MACs across all datasets confirms the effectiveness of our rank-constrained parameterization compared to quadratic $\mathcal{O}(r^2)$ complexity of full spectral filtering methods. Remarkably, rank-2 approximations achieve 6–12× parameter reduction compared to rank-24 while maintaining comparable accuracy across all tested configurations. This efficiency gain is particularly pronounced in High-Dimensional datasets where the Traffic dataset maintains optimal performance with minimal computational overhead.

**Forecast Horizon Consistency:** Comparing performance across different forecast horizons (336 vs. 720) reveals that spectral rank sensitivity remains consistent regardless of prediction length. This temporal invariance suggests that the low-dimensional spectral structure captures fundamental periodicities that persist across different forecasting contexts, validating our frequency domain processing approach for both short-term and long-term predictions.

**Spectral Sparsity Confirmation:** The universal performance saturation observed beyond rank-4 to rank-8 across all datasets provides strong empirical validation for the spectral sparsity hypothesis underlying our frequency domain pathway. This confirms that time series data naturally exhibit low-rank spectral structure, where a small number of dominant frequency components capture the majority of temporal dynamics. The minimal performance degradation at extremely low ranks (rank-2) demonstrates the effectiveness of our adaptive spectral filtering in identifying and preserving the most informative frequency components while discarding redundant spectral information.

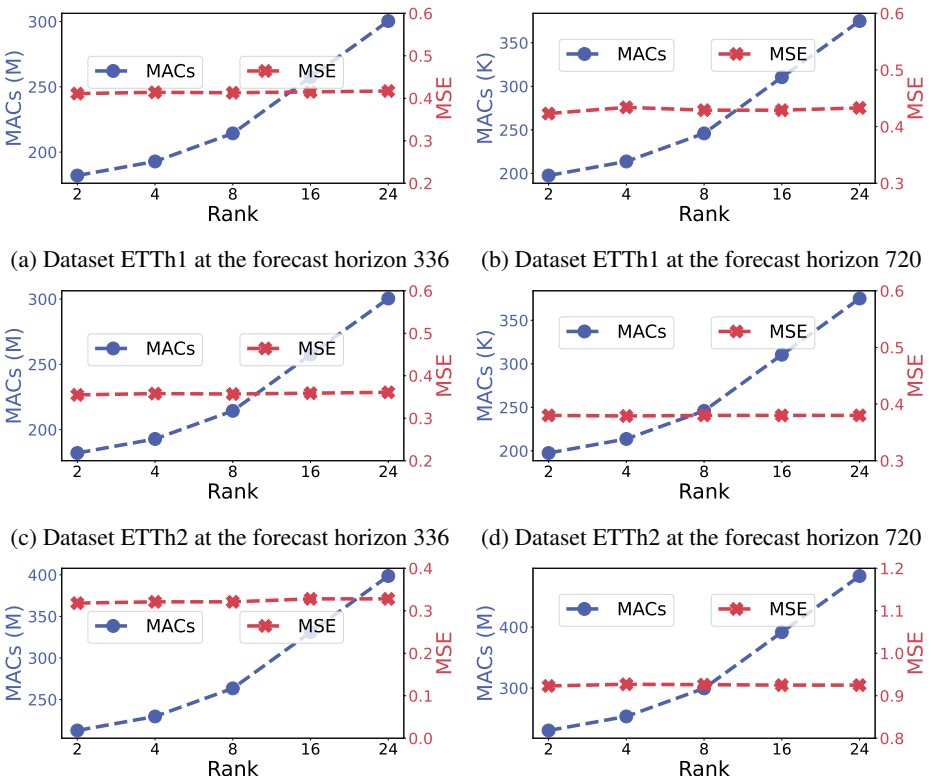

(a) Dataset ETTh1 at the forecast horizon 336    (b) Dataset ETTh1 at the forecast horizon 720

(c) Dataset ETTh2 at the forecast horizon 336    (d) Dataset ETTh2 at the forecast horizon 720

(e) Dataset Exchange at the forecast horizon 336 (f) Dataset Exchange at the forecast horizon 720

Figure 10: Performance of MixLinear with different spectral ranks on low-dimensional datasets.

## C.4 EXPERIMENTAL RESULTS ON EFFICIENCY

We evaluate the computational efficiency of the models by measuring MACs under different horizons. As Figure 12 shows, MixLinear consistently demonstrates the lowest computational overhead and the most stable growth rate under all tested scenarios. This best scalability is particularly evident in high-dimensional, long-horizon tasks ($H = 336$ and $H = 720$). In these regimes, MixLinear achieves up to a 40% reduction in MACs compared to SparseTSF. The more flat curve confirms MixLinear's time complexity of $\mathcal{O}(n \log n)$. This contrasts sharply with the significantly steeper growth rates observed for SparseTSF and FITS. In low-dimensional datasets with few channels (e.g., ETTh1 with 7 variables), MixLinear attains the lowest MAC cost—196.56K at horizon 720—outperforming SparseTSF (277.20K, +41.32%) and FITS (292.32K, +48.98%). This efficiency persists in high-dimensional settings: on the Traffic dataset with 862 channels, MixLinear again achieves the smallest MAC at 24.2M for horizon 720, compared to 34.14M for SparseTSF (+41.67%) and 36.00M for FITS (+48.76%). Overall, the experimental results unequivocally confirm that MixLinear substantially reduces computational overhead across various datasets and forecasting horizons while maintaining competitive predictive performance.

## C.5 EFFECT OF DOWNSAMPLING FACTOR ON MIXLINEAR PERFORMANCE

To evaluate the effect of the downsampling factor $\pi$ on model performance, we perform ablation studies on both low-dimensional (ETTh2) and high-dimensional (Solar) datasets. Table 6 demonstrates that MixLinear achieves optimal performance at $\pi = 24$ across most configurations, while maintaining robust forecasting accuracy across a wide range of downsampling factors. The results reveal several key insights: First, minimal downsampling ($\pi = 2$) does not yield the best performance, suggesting that moderate temporal aggregation helps the model capture more robust patterns by reducing noise and focusing on essential trends. Second, the optimal value $\pi = 24$ strikes an effective balance between computational efficiency and information preservation, achieving the lowest MSE values of 0.282, 0.336, 0.356, and 0.380 for ETTh2 across horizons 96, 192, 336, and 720 respectively.

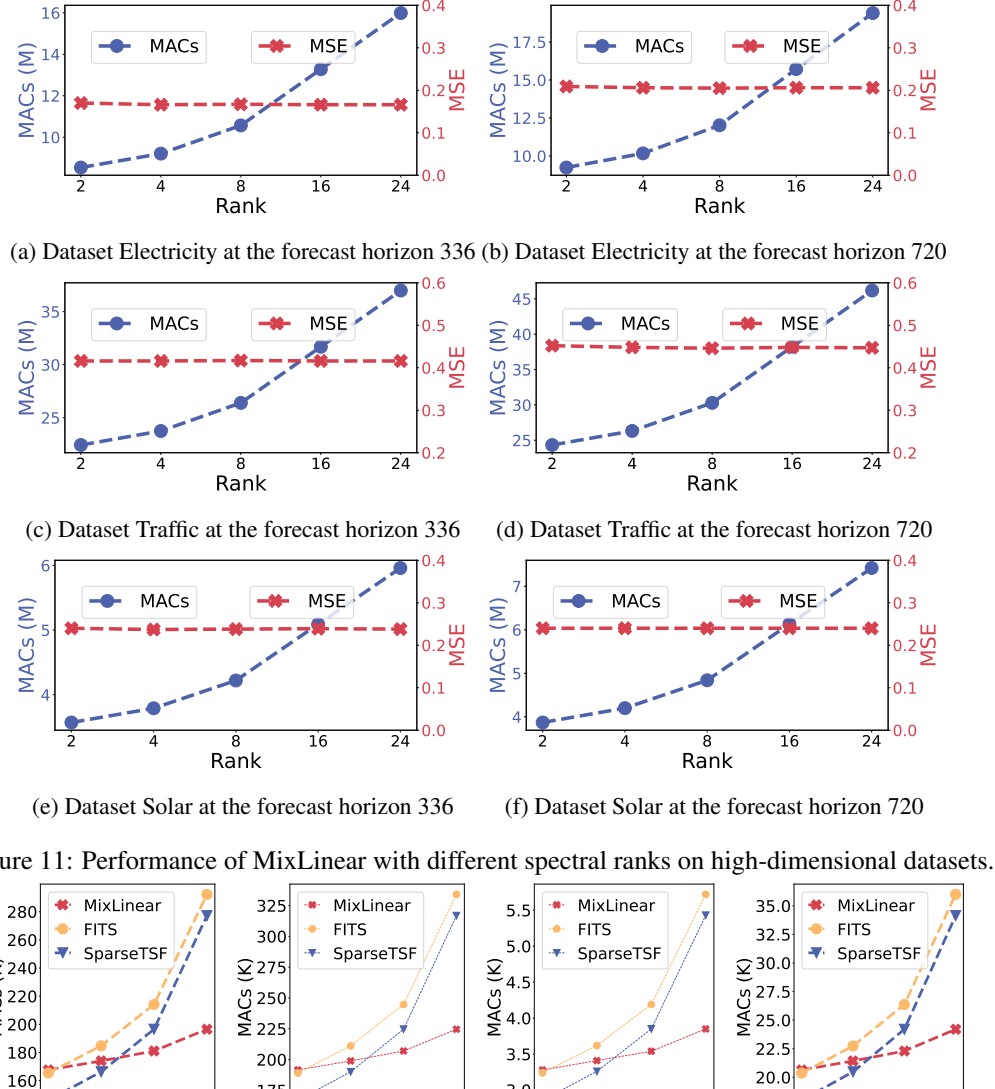

(a) Dataset Electricity at the forecast horizon 336 (b) Dataset Electricity at the forecast horizon 720

(c) Dataset Traffic at the forecast horizon 336 (d) Dataset Traffic at the forecast horizon 720

(e) Dataset Solar at the forecast horizon 336 (f) Dataset Solar at the forecast horizon 720

Figure 11: Performance of MixLinear with different spectral ranks on high-dimensional datasets.

(a) ETT (Low Channel) (b) Excahnge (Low Channel) (c) Solar (High Channel) (d) Traffic (High Channel)

Figure 12: Comparisons on MACs across different forecast horizons.

Third, excessive downsampling ($\pi = 36$) leads to performance degradation, indicating information loss when compression becomes too aggressive. Notably, the performance variations across different $\pi$ values remain relatively small (typically within 2-3% MSE difference), demonstrating MixLinear's inherent robustness to the choice of downsampling factor. This stability across different downsampling rates makes MixLinear adaptable to various computational budgets and deployment scenarios without requiring extensive hyperparameter tuning.

Table 6: MSEs of MixLinear with different downsampling factor $\pi$ on ETTh2 and Solar datasets. Bold values indicate best performance for each horizon.

| Dataset | Horizon | $\pi = 2$ | $\pi = 4$ | $\pi = 8$ | $\pi = 16$ | $\pi = 24$ | $\pi = 36$ |
|---|---|---|---|---|---|---|---|
| ETTh2 | 96 | 0.287 | 0.286 | 0.284 | 0.289 | **0.282** | 0.305 |
| | 192 | 0.351 | 0.358 | 0.341 | 0.360 | **0.336** | 0.348 |
| | 336 | 0.366 | 0.364 | 0.365 | 0.364 | **0.356** | 0.370 |
| | 720 | 0.390 | 0.383 | 0.383 | 0.389 | **0.380** | 0.391 |
| Solar | 96 | 0.207 | **0.205** | 0.215 | 0.209 | 0.211 | 0.218 |
| | 192 | 0.231 | 0.233 | 0.238 | 0.238 | **0.227** | 0.230 |
| | 336 | 0.250 | 0.254 | 0.258 | 0.251 | **0.240** | 0.242 |
| | 720 | 0.252 | 0.250 | 0.259 | 0.252 | **0.240** | 0.243 |

## D  VISUALIZATION

To showcase the prediction performance of MixLinear and compare it with other models, we present visualizations of their prediction results. Figures 13 and Figures 14 display the prediction results on the Exchange dataset for different models under two settings: input-720-predict-96 (Figure 13) and input-720-predict-192 (Figure 14). In these figures, the blue lines represent the ground truth values, while the orange lines denote the model predictions.

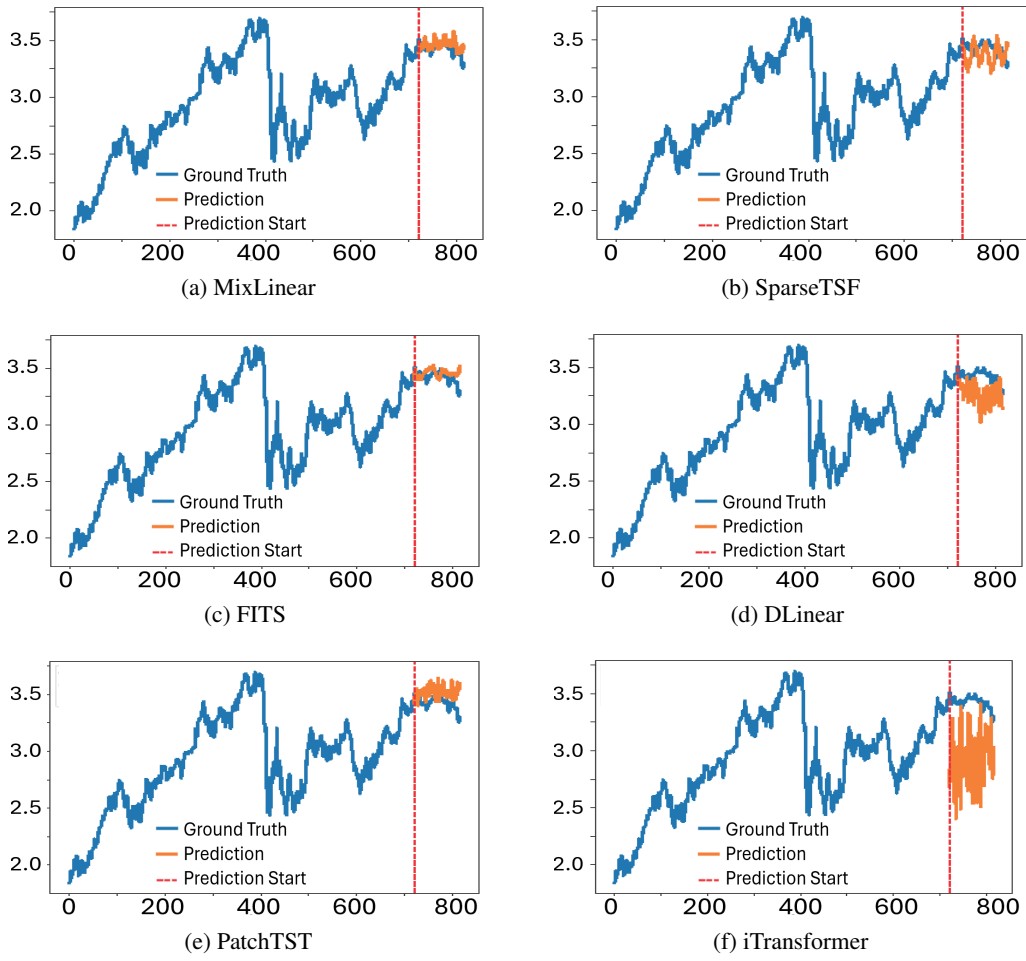

Figure 13: Prediction cases from Exchange by different models under the input-720-predict-96 settings. Blue lines are the ground truths, and orange lines are the model predictions.

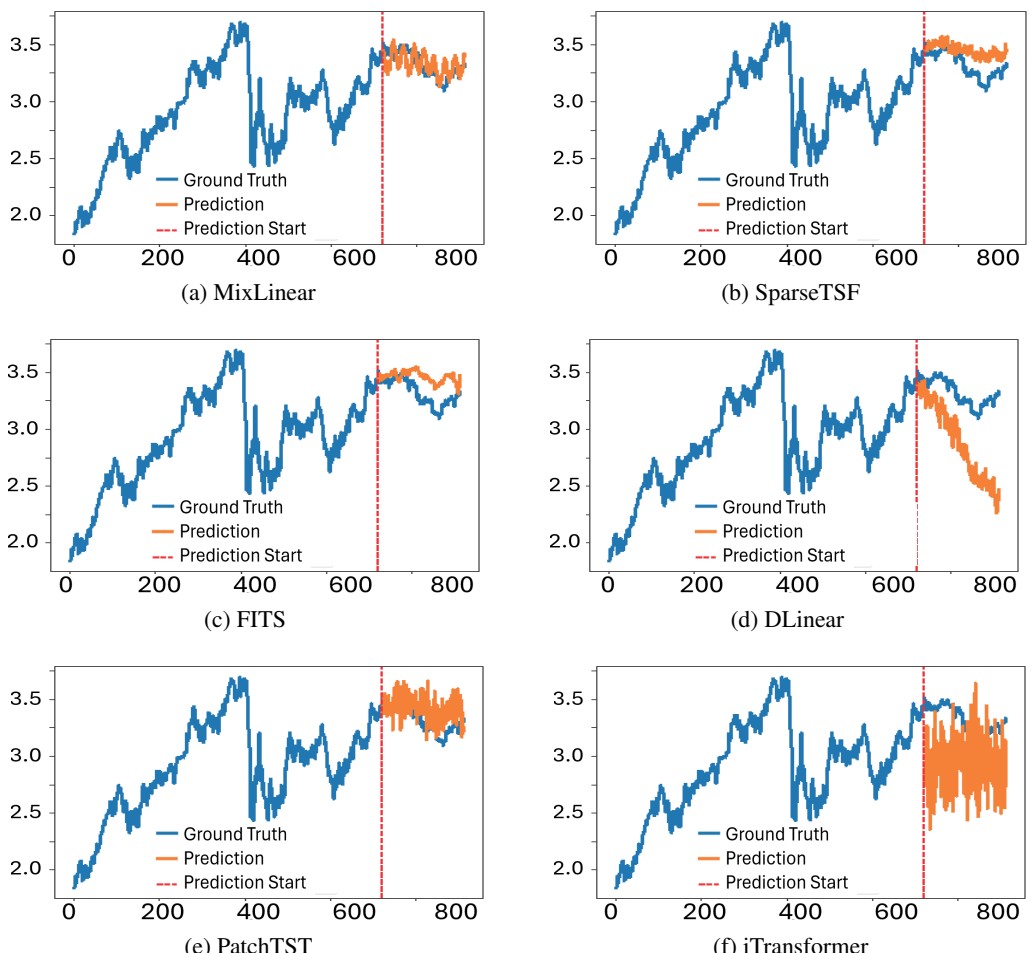

Figure 14: Prediction cases from Exchange by different models under the input-720-predict-192 settings. Blue lines are the ground truths and orange lines are the model predictions.

In Figure 13, the models predict 96 future time steps based on 720 past time steps. MixLinear shows strong alignment with the ground truth, capturing short-term patterns effectively. SparseTSF performs reasonably well, though slight deviations are observed. FITS and PatchTST closely follow the ground truth, demonstrating robust short-term forecasting capabilities. DLinear and iTransformer, however, exhibit larger deviations, indicating less accuracy for short-term predictions.

In Figure 14, the models are tasked with predicting 192 future time steps using 720 past time steps. MixLinear continues to perform accurately with minimal deviations, proving its effectiveness for longer prediction horizons. SparseTSF and FITS display moderate accuracy but show occasional mismatches in trends. PatchTST maintains strong performance, similar to the 96-step setting, while DLinear and iTransformer show greater discrepancies and instability, struggling with the extended horizon.

Overall, these figures highlight the strengths and weaknesses of the models. MixLinear and PatchTST consistently deliver accurate predictions across both settings, whereas DLinear and iTransformer face challenges in capturing longer-term temporal patterns. This comparison underscores the importance of robust model design for both short-term and long-term forecasting.

# E    USE OF LARGE LANGUAGE MODELS

ChatGPT was used as a general-purpose writing assistance tool to improve the grammar and clarity of writing during the preparation of this paper. LLMs did not contribute to the formulation of the

research idea, the experimental design, the data analysis, or the formulation of scientific conclusions. The authors assume full responsibility for all contents presented in this paper.

