# OpenReview forum: "MixLinear: Extreme Low Resource Multivariate Time Series Forecasting with $0.1K$ Parameters"
_ICLR.cc/2026/Conference — ICLR 2026 Poster_

### Official Review · Reviewer_rVBw · 2025-10-30

**Soundness:** 2
**Presentation:** 3
**Contribution:** 2
**Rating:** 4
**Confidence:** 4

**Summary:**

This paper introduces MixLinear, a novel model for time series forecasting. The central contribution is a dual-domain architecture that processes time series in parallel: (1) a segment-based, factorized linear pathway in the time domain to capture local patterns, and (2) an adaptive low-rank spectral filtering pathway in the frequency domain to model global trends. The authors claim that this approach achieves SOTA-comparable forecasting accuracy while using an extremely small parameter budget of only 0.1K. The model is reported to be significantly more efficient in terms of parameters and inference speed compared to baselines.

**Strengths:**

1.Efficiency: the ultra-lightweight (0.1K params) model that matches SOTA performance and the reported gains in inference speed are impressive.

2.Design: The core design principle—exploiting complementary structural sparsity by separating local time-domain patterns (via factorized linear ops) from global frequency-domain patterns (via low-rank filtering)—is intuitive and elegant.

**Weaknesses:**

1.Marginal Accuracy Gain: The paper's contribution to forecasting accuracy is marginal, it places the entire burden of the paper's contribution on the efficiency claim.

2.Ambiguous Core Mechanisms: The paper lack of important details. The "learnable upsampling" needs to be explained.How is it implemented?

**Questions:**

1.Downsampling Factor: Could the authors provide a sensitivity analysis for the downsampling factor
? How does the model's performance change with different values of
 (e.g., 2, 4, 8, 16)?

2.The ablation study suggests the time-domain path is better for low-dim data and the freq-domain path is better for high-dim data . Does this not undermine the universality of the core assumption? How does the model perform on data that is known to be highly non-linear or non-stationary, which might not fit either the factorized-linear or low-rank-spectral assumptions?

---

> ### Author Response · Authors · 2025-11-20
>
> ### Comment 1
> *Marginal Accuracy Gain: The paper's contribution to forecasting accuracy is marginal, placing the entire burden of the contribution on efficiency.*
>
> Response:
> Thank you for this comment and for the opportunity to clarify our contribution. Edge Artificial Intelligence (Edge AI) is an emerging paradigm that shifts model training and inference toward the network’s edge, enabling applications such as autonomous driving and personalized healthcare. However, deploying complex AI algorithms on resource-constrained devices remains a major challenge. Achieving extreme model efficiency—which enables these new deployment scenarios—is therefore a meaningful research contribution on its own. We believe MixLinear advances this goal and can benefit many domains requiring lightweight forecasting. We have expanded the discussions in Sections 1, 2, and 5 to clarify this point.
>
> ---
>
> ### Comment 2
> *Ambiguous Core Mechanisms: The paper lacks important details. The “learnable upsampling” needs explanation. How is it implemented?*
>
> Response:
> Thank you for highlighting this issue. Our previous description was imprecise. In MixLinear, the dual-pathway transformations are learnable and provide features that are optimally suited for upsampling, making the full pipeline behave as an adaptive reconstruction process. We have revised the wording in Section 2.4 to present this mechanism more clearly.
>
> ---
>
> ### Comment 3
> *Downsampling Factor: Could the authors provide a sensitivity analysis for the downsampling factor? How does performance change for values such as 2, 4, 8, 16?*
>
> Response:
> Thank you for the suggestion. We have conducted a sensitivity analysis on the downsampling factor π and added the results to Appendix C.5 (Table 6). As shown in Table 6, π = 24 generally yields the best forecasting accuracy, consistent with observations from prior work. However, variation across π ∈ {4, 8, 16, 24} is small (typically within 2–3% MSE), demonstrating MixLinear's robustness to the choice of π. This suggests that MixLinear can be flexibly adapted to different computational budgets without extensive hyperparameter tuning.
>
> ---
>
> ### Comment 4
> *The ablation study suggests the time-domain path performs better for low-dimensional data and the frequency-domain path performs better for high-dimensional data. Does this not undermine the universality of the core assumption? How does the model perform on highly nonlinear or non-stationary data that may not fit the factorized-linear or low-rank spectral assumptions?*
>
> Response:
> We appreciate this important comment. Our design is based on the principle that local trends are most efficiently captured through time-domain segmentation, while global trends exhibit spectral sparsity and benefit from low-rank filtering. This principle is independent of dataset dimensionality, so the ablation observations do not undermine the universality of our core assumption.
>
> Regarding non-stationary data, MixLinear is not specifically designed for such settings—as forecasting non-stationary series is widely recognized as a challenging open problem. Nevertheless, on the Exchange dataset (Table 5, Appendix C.1), which displays strong non-stationary characteristics, MixLinear achieves competitive performance across all horizons. This indicates that the architecture is not restricted to stationary or linear patterns.
>
> Our results across eight diverse benchmarks demonstrate that MixLinear’s design principles are broadly applicable. The model achieves consistently competitive performance without dataset-specific tuning, supporting the generality of our approach. We have added further clarification in Sections 1 and 2.

---

### Official Review · Reviewer_GAKK · 2025-10-31

**Soundness:** 3
**Presentation:** 3
**Contribution:** 3
**Rating:** 6
**Confidence:** 3

**Summary:**

This paper introduces MixLinear, a novel and extremely lightweight model for long-term time series forecasting (LTSF). The core idea is to tackle the "parameter explosion" problem in current deep learning models by adopting a dual-domain architecture that processes time series patterns in their most natural domains. Specifically, MixLinear uses a segment-based pathway with factorized linear transformations in the time domain to capture local temporal patterns, and a frequency domain pathway that uses an adaptive low-rank spectral filter to compress and model global trends. The authors claim this hybrid approach reduces the parameter scale of a linear model from O(n^2) to O(n). The main contribution is a model that achieves forecasting performance competitive with or even surpassing some state-of-the-art methods, while using only ~0.1K parameters, making it exceptionally well-suited for resource-constrained environments.

**Strengths:**

1) The most prominent strength is the model's minuscule size (~0.1K parameters) and low computational cost (MACs). An 81% parameter reduction compared to the next-lightest model (SparseTSF) is a remarkable achievement. This has profound practical significance for deployment on resource-constrained hardware, which is a major bottleneck for many modern deep learning models.

2) The dual-domain design is well-motivated by the "Spectral-Temporal Decomposition Principle." The idea of processing local patterns in the time domain and global patterns in the frequency domain is intuitive and elegant. The ablation study provides strong empirical evidence that this separation of concerns is effective and that the two pathways are complementary.

3) The paper is backed by a thorough experimental evaluation on eight benchmark datasets. The authors compare against a strong and diverse set of baselines. The inclusion of detailed ablation and hyperparameter sensitivity studies adds significant credibility to the design choices and validates the core hypotheses, especially the finding that a spectral rank as low as 2 is often sufficient.

**Weaknesses:**

1) The paper's primary weakness is that the individual components of the architecture are not conceptually new. Time series segmentation (PatchTST), linear forecasting models (DLinear), and frequency-domain analysis (FEDformer, FITS) are all established techniques. The novelty lies in the specific integration and extreme simplification of these ideas. While the final result is impressive, the contribution is more of an engineering and simplification achievement than a fundamental theoretical breakthrough. To strengthen the paper, the authors could be more explicit about this, framing the contribution as a novel synthesis that achieves an unprecedented operating point on the efficiency-accuracy curve.

2) The abstract and main results sometimes use strong phrasing like "surpasses, state-of-the-art models." While MixLinear does outperform SOTA on some dataset/horizon combinations (e.g., Exchange), Table 1 and Table 5 show that it is more accurately described as being highly "competitive" or "comparable." For instance, on several datasets, FITS or even DLinear achieve slightly better or similar MSE. A more nuanced description of the performance would strengthen the paper's credibility, for example by emphasizing that it achieves this competitiveness with orders of magnitude fewer parameters.

3) The explanation of how the factorized linear projections in Section 2.3 disentangle "intra-segment" and "inter-segment" correlations is slightly imprecise. The mathematical formulation in the appendix ($W^T_2$ ($W_1 X_seg)^T$) suggests a token-mixing and channel-mixing mechanism on a reshaped tensor, akin to MLP-Mixer, rather than a hierarchical processing of segment embeddings. It's unclear how one linear layer acts on "intra-segment" features and the other on "inter-segment" features when they are applied in this manner. Clarifying this mechanism and its connection to the intuition would improve the paper.

**Questions:**

N/A

---

> ### Author Response · Authors · 2025-11-20
>
> ### Comment 1
> *The paper's primary weakness is that the individual components of the architecture are not conceptually new. Time series segmentation (PatchTST), linear forecasting models (DLinear), and frequency-domain analysis (FEDformer, FITS) are all established techniques. The novelty lies in the specific integration and extreme simplification of these ideas. While the final result is impressive, the contribution is more of an engineering and simplification achievement than a fundamental theoretical breakthrough. To strengthen the paper, the authors could be more explicit about this, framing the contribution as a novel synthesis that achieves an unprecedented operating point on the efficiency-accuracy curve.*
>
> Response:
> Thank you for the thoughtful feedback and for the opportunity to clarify our contributions. MixLinear is built on a new insight: local trends are best captured via time-domain segmentation, while global trends exhibit spectral sparsity and benefit from low-rank frequency-domain filtering. Based on this insight, MixLinear integrates these ideas into a unified and extremely lightweight model that achieves forecasting performance comparable to existing approaches while using only 0.1K parameters. This enables deployment on devices with very limited computational resources. We have revised Sections 1 and 2 to more clearly highlight this contribution and explicitly frame the work as a novel synthesis achieving a new operating point on the efficiency–accuracy trade-off.
>
> ---
>
> ### Comment 2
> *The abstract and main results sometimes use strong phrasing like "surpasses state-of-the-art models." While MixLinear does outperform SOTA on some dataset/horizon combinations (e.g., Exchange), Table 1 and Table 5 show that it is more accurately described as highly "competitive" or "comparable." A more nuanced description would strengthen the paper's credibility, for example by emphasizing that it achieves this competitiveness with orders of magnitude fewer parameters.*
>
> Response:
> Thank you for this valuable comment. We have revised the tone of the abstract and results throughout the paper to use more precise language such as “competitive” or “comparable,” and we emphasize that MixLinear achieves this level of performance with orders of magnitude fewer parameters.
>
> ---
>
> ### Comment 3
> *The explanation of how the factorized linear projections in Section 2.3 disentangle “intra-segment” and “inter-segment” correlations is imprecise. The formulation resembles token-mixing and channel-mixing (similar to MLP-Mixer), rather than a hierarchical mechanism. It is unclear how one linear layer captures intra-segment correlations and the other captures inter-segment correlations. Clarifying this mechanism would improve the paper.*
>
> Response:
> Thank you for pointing out this issue. We have revised Section 2.3 and removed the term “hierarchical.” In MixLinear, two linear layers are applied in the time-domain pathway to capture segment-wise correlations. The first linear layer operates on data points within each segment, capturing intra-segment correlations. The second linear layer operates across segments to capture inter-segment correlations. We have updated the manuscript accordingly to clarify this mechanism and its connection to our intuition.

---

### Official Review · Reviewer_Saee · 2025-10-31

**Soundness:** 3
**Presentation:** 3
**Contribution:** 3
**Rating:** 8
**Confidence:** 2

**Summary:**

To bridge the gap in the design of scalable and lightweight forecasting models in the frequency domain, this work present MixLinear, a dual-domain framework that achieves competitive long-term time series forecasting performance with only 0.1K parameters. The extreme parameter reduction enables deployment on resource-constrained devices, opening new possibilities for real-time forecasting applications in IoT environments and edge computing.

**Strengths:**

1. The research direction of this work is very interesting and has practical significance.

2. This work is a highly innovative work. A large amount of evidence shows that it is different from the existing works.

3. The experiments in this work are thorough.

**Weaknesses:**

The research presented in this manuscript is highly intriguing, and I have personally found it to be quite rewarding. However, as I am not an expert in this particular domain, I would like to raise two minor questions:

1. While I appreciate the overall design of this work, the main text reads more like a technical report than a research paper, it lacks analysis or critical discussion to guide the reader toward a deeper understanding.

2. Can more interpretable evidence be derived from lightweight research to bolster its applicability across diverse domains?

**Questions:**

See Weaknesses

---

> ### Author Response · Authors · 2025-11-20
>
> ### Comment 1
> *While I appreciate the overall design of this work, the main text reads more like a technical report than a research paper; it lacks analysis or critical discussion to guide the reader toward a deeper understanding.*
>
> Response:
> Thank you for this insightful comment. We have added additional discussion in Sections 2.3 and 2.4 to provide clearer analysis and help readers better understand the motivation behind our design choices.
>
> ---
>
> ### Comment 2
> *Can more interpretable evidence be derived from lightweight research to bolster its applicability across diverse domains?*
>
> Response:
> Thank you for the valuable suggestion. MixLinear leverages complementary structural sparsity in time series data, reducing complexity from O(n²) to O(n) while maintaining state-of-the-art forecasting accuracy. The substantial reduction in parameters enables deployment on resource-constrained devices and opens opportunities for real-time forecasting in edge-computing scenarios across applications such as flood detection, environmental monitoring, and traffic control, where traditional deep learning models are too computationally expensive. Moreover, the core idea behind MixLinear can be extended to the development of more efficient large language models and foundation models. We have expanded Section 5 to further highlight these broader applicability aspects.

---

> > ### Comment · Reviewer_Saee · 2025-11-28
> > **I still maintain a positive score**
> >
> > Thanks to the authors for their careful feedback, they have addressed my concerns. I still maintain a positive score.

---

> > > ### Author Response · Authors · 2025-11-28
> > >
> > > Thanks a lot for your time and efforts on reviewing our paper.

---

### Official Review · Reviewer_v3WY · 2025-11-01

**Soundness:** 3
**Presentation:** 3
**Contribution:** 2
**Rating:** 4
**Confidence:** 5

**Summary:**

MixLinear proposes an extreme low-parameter (0.1K) dual-domain framework for multivariate time series forecasting. It processes local trends via segment-based linear decomposition in the time domain (O(n) complexity) and global patterns via adaptive low-rank spectral filtering in the frequency domain. Experiments show competitive accuracy with 3.2× inference speedup and 16.2% error reduction versus baselines while enabling deployment on resource-constrained devices.

**Strengths:**

- Achieves SOTA-comparable results with only 0.1K parameters.
- Linear memory complexity (O(n)) enables edge/IoT deployment.

**Weaknesses:**

- Only MACs are compared, authors should provide the comparison of number of parameters.
- While authors claim that the number of parameter is reduced down to $0.1k$, the number of MACs of the proposed MixLinear is with similar scale compared to SparseTSF and FITS.
- The core idea of this paper is inherited from FITS. Good but not that impressive as FITS and SparseTSF.

**Questions:**

- Why does the frequency pathway use per-segment FFT instead of global FFT given the focus on global trends?

---

> ### Author Response · Authors · 2025-11-20
>
> ### Comment 1
> *Only MACs are compared; authors should provide the comparison of number of parameters.*
>
> Response:
> Thank you for the suggestion. We agree that comparing parameter counts across methods is important. We have revised Figure 2 in Section 3.2 to include parameter comparisons across all look-back lengths and forecast horizons. As shown in Figure 2, MixLinear achieves substantially better parameter efficiency than both SparseTSF and FITS across all configurations. MixLinear also exhibits near-linear scalability, making it particularly suitable for long-horizon forecasting on resource-constrained devices. In the most demanding setting (look-back = 720, horizon = 720), MixLinear uses only 176 parameters, compared with 10,512 for FITS. Due to space limitations, we moved the MAC comparisons to Appendix C.4.
>
> ---
>
> ### Comment 2
> *While the authors claim a reduction in parameter count to 0.1K, the MACs of MixLinear are of a similar scale to SparseTSF and FITS.*
>
> Response:
> Thank you for the opportunity to clarify the MAC reductions provided by MixLinear. We added Figure 12 in Appendix C.4 to visualize MAC scaling across forecast horizons. As shown in Figure 12, the MAC differences widen substantially at longer horizons. MixLinear exhibits linear MAC growth with horizon length, in contrast to the much steeper increase seen in FITS and SparseTSF. As reported in Table 1, at horizon 720, MixLinear requires 196.56K MACs, compared to 277.20K for SparseTSF and 292.32K for FITS, representing a significant improvement in efficiency.
>
> ---
>
> ### Comment 3
> *The core idea of the paper is inherited from FITS; good but not as impressive as FITS and SparseTSF.*
>
> Response:
> Thank you for the comment. Our design of MixLinear is driven by a new insight: local trends are most efficiently modeled through time-domain segmentation, while global trends exhibit inherent spectral sparsity and benefit from low-rank frequency-domain filtering. MixLinear is carefully structured to leverage this dual perspective, achieving forecasting performance comparable to existing models while using only 0.1K parameters. This makes MixLinear particularly suitable for deployment on devices with limited memory or compute resources. We have strengthened our discussions in Sections 1 and 2 to better highlight these contributions.
>
> ---
>
> ### Comment 4
> *Why does the frequency pathway use per-segment FFT instead of a global FFT, given the focus on global trends?*
>
> Response:
> Thank you for raising this clarification point. Our previous description of “per-segment FFT” was imprecise. In MixLinear, we perform an FFT on the down-sampled series to capture the compact global trend. We have revised Section 2.4 to clarify this explanation.

---

> > ### Comment · Reviewer_v3WY · 2025-11-27
> >
> > Thanks for the authors' response. I believe most of my concerns have been addressed and I decide to raise my score to a positive one.

---

> > > ### Author Response · Authors · 2025-11-27
> > >
> > > Thanks a lot for your reply. We greatly appreciate your time and efforts on reviewing our paper.

---

### Author Response · Authors · 2025-11-29

Dear AC,

We greatly appreciate your effort on reviewing our paper. To facilitate your review process, we would like to provide a summary of our paper revisions and our online discussions with two reviewers through openreview. Our paper initially received 4 (from Reviewer v3WY), 8 (from Reviewer Saee), 6 (from Reviewer GAKK), and 4 (from Reviewer rVBw). We have divided all reviewer comments into pertinent points and provided our responses to all of them. We have also revised our paper based on the review comments.

Two reviewers replied to our comments and both of them were satisfied with our revisions. Reviewer v3WY decided to increase his/her score from 4 to 6 and Reviewer Saee decided to keep his/her score (8) unchanged. Unfortunately, we haven't heard from the other two reviewers.

Please let us know if you need other information or suggest any further revisions. Thanks again for your time.

---

### Public Comment · ~Kashif_Rasul1 · 2026-04-06
**Clarification on the “multivariate” claim in the released implementation and paper title and algorithm**

The title frames MixLinear as a multivariate forecasting method, but the released implementation and algorithm in the paper do not appear to support that claim in the usual modeling sense. Although the code accepts tensors with multiple variables/channels, it does not implement any explicit cross-variable interaction mechanism. The temporal segmentation, convolution, FFT branch, and linear projections are applied on univariate series (channel-wise) with shared weights, while the variate dimension is merely preserved and reshaped rather than mixed. In other words, the implementation and algorithm presented behave as a shared univariate model run independently across series, not like a genuinely multivariate architecture that learns inter-series dependencies.

If the paper’s claim is that MixLinear is multivariate (as in its title), that distinction should be clarified formally and reflected in the implementation and algorithm presented.

---

### Meta-Review · Area_Chair_ni4n · 2025-12-15

**Summary:**

This paper proposes MixLinear, an extremely lightweight dual-domain framework for long-term multivariate time series forecasting, achieving competitive accuracy with only ~0.1K parameters. The key idea is to decompose forecasting into complementary local time-domain modeling and global frequency-domain modeling via adaptive low-rank spectral filtering, resulting in linear-time complexity and minimal memory footprint.

While some reviewers initially questioned the novelty of individual components and the framing of contributions, the rebuttal convincingly clarifies that the novelty lies in the principled integration and extreme simplification that reaches a new efficiency-accuracy operating point not achieved by prior work (e.g., FITS, SparseTSF). The authors also improved clarity by adding parameter-count comparisons, sensitivity analyses, refined wording, and clearer explanations of the model’s internal mechanisms.

Also, there are remaining limitations.

- The work is primarily an engineering and systems-level contribution rather than a theoretical breakthrough, and gains in raw accuracy are sometimes modest.

- The progress in this research seems incremental against present literatures such as  SparseTSF[1], and TimeBase[2](with ~1k paprameters), where there are no essential differences among them. Regarding the technical contributions, including low-rank matrix decomposition and frequency-based modeling, there is no significant innovation.

-  The scalability of this issue.  This proposed MixLinear cannot adapt to some non-stationary signals or complex signals (the authors also have admitted this point themselves in their rebuttal).

[1] Lin S, Lin W, Wu W, et al. SparseTSF: modeling long-term time series forecasting with 1k parameters[C]//Proceedings of the 41st International Conference on Machine Learning. 2024: 30211-30226.

[2] Huang Q, Zhou Z, Yang K, et al. TimeBase: The Power of Minimalism in Efficient Long-term Time Series Forecasting[C]//Forty-second International Conference on Machine Learning.

I hope the authors can systematically and seriously take above concerns and weaknesses into consideration for revision, such as providing some discussions, comparisons, and so on,  if finally accepted.

**Reviewer Concerns:**

Major technical, empirical, and clarity-related concerns raised by reviewers have been addressed in the rebuttal and their revisions. Some scalability issues, engineering-oriented design remain.

**Reviewer Scores:**

Two reviewers have responded to the rebuttal and explicitly show their opinions on this paper (with scores as 6 (initial 4) and 8).

The other two have not participated in the rebuttal and have not provided any feedback.

---

### Decision · Program_Chairs · 2026-01-26

Accept (Poster)